# Very high-resolution modelling of submesoscale turbulent patterns and processes in the Baltic Sea

Reiner Onken<sup>1</sup>, Burkard Baschek<sup>1</sup>, and Ingrid M. Angel-Benavides<sup>1</sup> <sup>1</sup>Helmholtz-Zentrum Geesthacht, Max-Planck-Straße 1, 21502 Geesthacht, Germany *Correspondence to:* Reiner.Onken@hzg.de

Abstract. In order to simulate submesoscale turbulent patterns and processes (STPPs) and to analyse their properties and dynamics, the Regional Ocean Modeling System (ROMS) is applied to a subregion of the Baltic Sea around the island of Bornholm. The modeled STPPs provide an aid for the interpretation of observations that were taken during the "Expedition Clockwork Ocean" in the same region in June 2016. To create a realistic mesoscale environment, ROMS with 500-m horizontal much time interpretation of the same region in June 2016.

- resolution is one-way nested into an existing operational model. The comparison of the results with satellite images shows good agreement. STPPs with horizontal scales < 1 km are resolved with a second nest of 100-m resolution providing a deep insight into ageostrophic processes and associated quantities when the atmospheric forcing is turned off. STPPs evolve rapidly within a about a day. They are characterised by strong vertical speeds of O(100) m day<sup>-1</sup> and relative vorticities and divergences reaching multiple of the Coriolis parameter. Typical elements of the secondary circulation of two-dimensional strain-induced
- frontogenesis are identified at an exemplary front in shallow water, and details of the ageostrophic flow field are revealed. The conditions for inertial and symmetric instability are evaluated for the whole domain, and the components of the tendency equation are computed in a subregion. While anticyclonic eddies are generated solely along coasts, cyclonic eddies are rolledup streamers and found in the entire domain. A special feature of the cyclones is their ability to absorb internal waves and to sustain patches of continous upwelling for several days favouring plankton growth. The kinematic properties show good
- agreement with observations, while some observed details within a small cyclonic eddy are only partly reproduced, most likely due to a lack of horizontal resolution or non-hydrostatic effects.

# 1 Introduction

An ambitious expedition was conducted in the Baltic Sea in June 2016 under the lead of the Helmholtz-Zentrum Geesthacht (HZG). The objective of the "Expedition Clockwork Ocean" was to observe submesoscale patterns and processes (STPPs) in order to better understand their role in the cascade of turbulent kinetic energy in the ocean, and to assess their impact on the primary production. In this article, high-resolution modeling is used to understand and interpret the observed features.

Submesoscale turbulence is characterised by horizontal scales between 10 m and 10 km, vertical scales from 1 to 100 m, and time scales from hours to days (McWilliams, 2016). Thus, in the turbulence spectrum of the ocean, the submesoscale occupies 4 orders of magnitude in the horizontal, 3 orders in the vertical, and 3 orders of magnitude in time. As the whole spectrum of

horizontal motions extends over 9 orders of magnitude between the basin-scale circulation ( $10^6$  m) and the dissipation scale ( $10^{-2}$  m, see Müller et al. (2005)), a significant fraction of the spectrum is occupied by the submesoscale. The submesoscale wavenumber band is the intermediate regime between the mesoscale and the microscale (McWilliams, 2008); it separates the larger-scale flow, where the Coriolis force dominates (Rossby number, Ro < 1), from the smaller scales where the rotation of

5 the Earth is of minor importance  $(Ro \sim O(1))$ .

The energy sources for ocean currents are mainly at the planetary scale (O(1000 km)), while the energy is removed at the microscale by viscous dissipation of kinetic energy where it is finally converted to heat. In order to equilibrate the ocean on climatological time scales, this demands a continous spectral flow from the planetary scale to the microscale, which is called the "blue" cascade. However, according to the theory of geostrophic turbulence (Charney, 1971; Rhines, 1979), the energy cascade

- 10 in low Rossby number regimes is "red", i.e. there is a feedback of energy from mesoscale turbulence to larger scales. This leads to the question how the mesoscale energy is dissipated. A generic answer was given by McWilliams (2016): "Submesoscale currents spontaneously emerge from mesoscale eddies and boundary currents, especially in the surface and bottom boundary layers. They are generated through instabilities, frontogenesis and topographic wakes" leading to the required forward cascade.
- The elements of submesoscale turbulence are essentially the same as in mesoscale turbulence: fronts, instabilities, meanders, eddies, and filaments. However, the submesoscale spatio-temporal scales are 1 or 2 orders of magnitude smaller. The first observational campaigns targeting explicitly at the characteristics of STPPs were the Lateral Mixing experiment (LatMix) in the North Atlantic (Shcherbina et al., 2015) and the Submesoscale Experiments (SubEx) in the Southern California Bight (Ohlmann et al., 2017).
- More detailed knowledge about the generation, kinematics, and dynamics of STPPs is conveyed by experiments with offline nested numerical models, so far predominantly applied to the regimes of eastern and western boundary currents. Using an idealised ROMS (Regional Ocean Modeling System, Shchepetkin and McWilliams (2005)) model with a flat bottom, a straight coast line, and uniform wind stress, Capet et al. (2008a, b, c) investigated the mesoscale to submesoscale transition in the California Current system. Based on 5 different model setups with variable horizontal resolution ranging from 12 to 0.75 km, they found that vigorous STPPs develop as the horizontal grid scale was reduced to O(1) km. The STPPs are initialised by the
- generation of submesoscale fronts that are strained between mesoscale eddies. Some of these fronts become unstable, develop submesoscale meanders, and fragment into roll-up vortices. These processes are accompanied by vertical velocities up to 50 m day<sup>-1</sup> and by relative vorticitiy values of O(f), where *f* is the Coriolis parameter. Computations of the energy balance showed that significant conversion from potential to kinetic energy (baroclinic instability) takes place, both in the mesoscale and submesoscale waveband. A significant forward energy cascade occurs in the submesoscale range in transit to dissipation
- at the microscale. Using ROMS at very high horizontal resolution (150 m) and with realistic topography, Gula et al. (2016a) studied the submesoscale structure in the interior of a mesoscale cyclonic eddy that was generated on the inshore side of the Gulf Stream interacting with the shelf slope. The model results revealed multiple submesoscale fronts at the rim of the eddy that appear as elongated vorticity filaments, reaching locally multiple of f. Like in the California Current System, these filaments become unstable and break up into strings of submesoscale vortices that are advected into the interior of the mesoscale eddy.
- There, the cyclonic vortices are associated with low-salinity patches, indicating upwelling. Furthermore, it was shown that the

divergence of the flow is dominated by intense small-scale patterns. These patterns are partially associated with the genesis of multiple fronts, but exhibit also characteristics of internal gravity waves propagating away from mesoscale fronts. In another article, Gula et al. (2014) investigated the life cycle of a submesoscale cold filament on the South Wall of the Gulf Stream: the formation of the filament (filamentogenesis) is primarily caused by an intensification of surface buoyancy gradients due to

5 horizontal straining flows and a two-celled ageostrophic secondary circulation with strong surface convergence and associated downwelling in the centre.

The theory of frontogenesis and the associated secondary circulation is significantly shaped by the meteorological literature (e.g. Eliassen (1962)) where a front is formed in the confluence zone of a deformation field (Fig. 9.1b in Holton (1982)), that is generated in the centre of two crosswise arranged pairs of cyclonic and anticyclonic eddies. Typical characteristics of such

- a front are an along-front geostrophic jet and a cross-front ageostrophic overturning cell with downwelling on the dense side and upwelling on the less dense side of the front. For mesoscale fronts, this picture was authenticated by the idealised model studies of Bleck et al. (1988), Nagai et al. (2006), and Thompson (2000). For submesoscale fronts, the very recent model study of McWilliams (2017) confirms that "These circulation patterns, ..., are qualitatively similar to those due to the 'classical' mechanism of strain-induced frontogenesis'.
- Baroclinic instability and instabilities driven by the horizontal shear of currents are the main processes leading to the instability of submesoscale fronts. The solutions of the classical problem of baroclinic instability (Charney, 1947; Eady, 1949) sufficiently explain the most unstable wavelength and growth rate of mesoscale disturbances, but these disturbances encompass a large fraction of the water depth including the surface layer and main thermocline. By contrast, another short-wave type of baroclinic instability was described by Blumen (1979) in the case of reduced stratification of the (atmospheric) boundary
- layer. The oceanic equivalents for this type are inertial (or centrifugal), symmetric, and mixed-layer instability. The latter was thoroughly investigated by Boccaletti et al. (2007) who showed that this class of instabilities may occur in the surface and bottom mixed layer; therefore, it is denoted as "mixed-layer instability". It is a hybrid instability composed of gravitational and baroclinic instabilities finally leading to a flattening of isopynals, and a positive (upward) buoyancy flux. It requires lateral buoyancy gradients, and it is composed of short modes between 200 m and 20 km and growth rates around 1 day<sup>-1</sup>. The start-
- ing point for such instabilities is the slumping of isopycnals that is constrained by rotation, because the thermal wind balance is established after a pendulum day. Thereafter, baroclinic instability generates wavelike disturbances that upset the thermal wind balance. As shown by Fox-Kemper and Ferrari (2008), the positive vertical buoyancy flux associated with mixed-layer instability is most important for the restratification of the upper ocean. By contrast, inertial (or centrifugal) instability does not require lateral buoyancy gradients but horizontal shear plus negative absolute vorticity. Therefore, inertial instability is driven
- by barotropic instability while a vertical buoyancy flux is not necessary. Pure inertial instability is unlikely to occur in the surface mixed layer, but it may arise when currents interact with topography on their right hand side (on the northern hemisphere), creating strong anticyclonic relative vorticity. A necessary condition for the occurrence of inertial instability is a relative vorticity smaller than -f (Haine and Marshall, 1998; Thomas et al., 2013). Symmetric instability is a hybrid gravitational-inertial instability, and may be considered to be a special type of mixed-layer instability. To a first approximation, it arises when the
- potential vorticity is negative and the Richardson number is Ri < 1 (Boccaletti et al., 2007). The numerical simulations of

Haine and Marshall (1998) confirm that symmetric instability dies as soon as Ri > 1 and baroclinic instability takes over. The transition from gravitational-inertial to gravitational-baroclinic instability was explored by Stamper and Taylor (2017) in detail. More in-depth conditions for symmetric instability were formulated by Taylor and Ferrari (2009) and Thomas et al. (2013).

Most of the numerical studies cited above manifest that STPPs act as a conveyer for kinetic energy from the mesoscale to the

5 microscale. Besides this role, STPPs are also relevant for primary production (Mahadevan, 2016). In particular, the associated strong upward vertical motions transport nutrients into the sunlit ocean where plankton growth occurs, while carbon fixed by the phytoplankton cells is removed from the euphotic layer by downwelling events (Lévy et al., 2012).

The "Expedition Clockwork Ocean" was conducted 20–28 June 2016. In order to reproduce a realistic mesoscale physical environment in the experimental area of the "Expedition Clockwork Ocean", a ROMS model with 500-m horizontal resolution

(referred to as R500 in the following) and realistic atmospheric forcing is nested into an existing coarser-resolution HBM operational model. An even finer nested ROMS model with a grid size of 100 m (R100) is used to enable the generation of STPPs under near-adiabatic conditions, thus providing a base for an analysis of their properties and dynamics.

The utilised numerical models are described in Section 2. The results of the numerical experiments are presented in Sections 3 and 4 and compared with observations in Section 5, followed by the conclusions. In the following, all time specifications

refer to the year 2016 (unless stated otherwise) and are given in UTC (Universal Time Coordinated).

## 2 The models

# 2.1 Geographic and oceanographic setting

The Baltic Sea is a semi-enclosed marginal shelf sea with a mean water depth of 55 m and with narrow shallow connections to the North Sea. Due to river runoff, the water balance is positive driving an estuarine circulation with quasi-permanent outflow

- of fresh surface waters and an intermittent inflow of salty water from the North Sea. At the surface, this creates a horizontal salinity gradient with high salinities in the west and almost freshwater conditions in the far north. Salinity is increasing with depth thus stratifying the water column year-round and generating a permanent halocline at about 60-m depth in the deeper basins. For more details see Feistel et al. (2008), Leppäranta and Myrberg (2009), or Osínski et al. (2010).
- The survey area of the "Expedition Clockwork Ocean" was located in the western Baltic, to the south of the island of 25 Bornholm (Fig. 1). This region is separated into two basins, the Arkona Basin and the Bornholm Basin. The Arkona Basin is the smaller one with a maximum water depth of 51 m, while the maximum depth of the Bornholm Basin is 92 m. The basins are connected by two channels, with an exchange of water limited by sills of 45 m depth in the Bornholmsgat (Magaard and Rheinheimer, 1974) and 31 m between Rönnebank and the island of Rügen.

## 2.2 HBM

HBM (HIROMB-BOOS model (Berg, 2012)) is an operational ocean circulation model providing forecasts of the physical conditions in the Baltic Sea. The horizontal resolution is about 1 nm (nautical mile). In the vertical direction, the model domain

is separated into 25 depth levels, spaced at 5 m between the sea surface and 100-m depth, and additional levels below at 150, 200, 300, and 400-m depth. The HBM forecasts are available at CMEMS (Copernicus Marine Environment Monitoring Service, http://marine.copernicus.eu, product BALTICSEA\_ANALYSIS\_FORECAST\_PHY\_003\_006, last access 1 March 2019) since 1 April 2015, comprising daily mean and hourly instantaneous fields. For this article, the HBM daily mean fields of June 2016

were utilised that contained 30 records of the prognostic variables potential temperature, salinity, horizontal velocity, and sea surface height for each June day.

## 2.3 ROMS

The employed numerical ocean circulation model ROMS is a hydrostatic, free-surface, primitive equations model. Its algorithms are described in detail in Shchepetkin and McWilliams (2005). In the vertical, the primitive equations are discretised

- over a variable topography using stretched terrain-following coordinates, so-called s-coordinates (Song and Haidvogel, 1994). In the horizontal, spherical coordinates are used. Biharmonic mixing along isopynic surfaces is applied to the tracers, both in R500 and R100, while biharmonic mixing of momentum is used in R500 and a monoharmonic formulation in R100. The vertical mixing of momentum and tracers is parameterised with the GLS (Generic Length Scale) scheme by Umlauf and Burchard (2003) in R500, and with the interior closure by Large et al. (1994) in R100, referred to as the KPP scheme. For the bottom
- friction, a quadratic law is applied, and the pressure gradient term is computed using the standard density Jacobian algorithm by Shchepetkin and Williams (2001). The air-sea interaction boundary layer in ROMS is formulated by means of the bulk parameterisation by Fairall et al. (1996). R500 and R100 have the parameters and equations listed in Table 1 in common, while the individual grid-size-dependent parameters and properties are summarised in Table 2. As the spatial scales of the smallest known STPPs are O(10 m), it is expected that large and medium-scale STPPs are resolved.

#### 20 2.4 Nesting, boundary conditions

There are two offline nesting steps: R500 is nested in HBM, and R100 is nested in R500. While the ROMS-to-ROMS nesting is technically straightforward, the first nest is somewhat more delicate, because the HBM output is provided on depth levels while ROMS uses s-coordinates. In addition, the bathymetry of HBM is not contained in the output provided by CMEMS. Therefore, the setup of the R500 domain and the nesting was accomplished as follows:

- 1. The bathymetry of the GEBCO\_2014 grid (General Bathymetric Chart of the Oceans, 30 arc seconds horizontal resolution) was used as the lower boundary of the R500 domain, and it was smoothed iteratively until the stability condition  $rx_0 \le 0.2$  was reached everywhere (see Haidvogel et al. (2000) and Table 2).
  - 2. The HBM prognostic variables were interpolated linearly onto the R500 horizontal grid at each HBM depth level in 24-hour intervals and for each day of June.
- 30 3. The R500 vertical grid was defined according to Table 1 and the HBM fields were interpolated vertically onto the R500 vertical grid. The first record of the resulting data set served as initial condition for the R500 integration, while the lateral boundary conditions were extracted from all records.

- 4. R500 was integrated for one day, and the near bottom velocities were checked for odd features that might have been caused by improper alignment of the R500 bathymetry with the unknown HBM bathymetry. If such features (e.g. abnormal vertical motions) were noticed, the R500 bathymetry was manually adjusted and the above procedure was iteratively repeated, starting with step 3.
- For the R500-to-R100 nesting, the same vertical grid definition was used, and no interpolation from depth levels to scoordinates was required. Special care was taken for the preparation of the initial and boundary conditions, as a nesting ratio of 5 is rather challenging: cubic splines were used for the horizontal interpolation of the prognostic variables of R500 onto the child's grid, because in preliminary test runs with linear interpolation, the relative vorticity of jet flows across the open boundaries looked weird because the width of cyclonic and anticyclonic shear zones was frequently the same. This is
- contrary to experience where the width of the cyclonic shear zone is narrower than that of the anticyclonic shear. Using spline interpolation, those jets became more realistic even though not perfect. In addition, the downscaled fields were generated in 3-hour intervals, leading to a smoother temporal change of the lateral boundary conditions. Cubic splines were used as well for the interpolation of the atmospheric forcing fields on the R100 grid. Because in a nested configuration, the s-coordinates of the parent and the child at any location are only identical if the bathymetry is the same, the bathymetry of R100 was cloned from
- R500 and linearly interpolated onto the finer grid.

For each nest, radiation boundary conditions with nudging (Marchesiello et al., 2001) were applied to temperature and salinity, barotropic and baroclinic momentum, and the mixing of turbulent kinetic energy along the lateral boundaries. The boundary conditions of the free surface were defined according to Chapman (1985). In all ROMS setups, the nudging time scales were set to 2 days for the corresponding variables. At the surface, the air-sea interaction in the ROMS nests was specified by means of atmospheric forcing fields from the ICON-EU model, provided by the German Weather Service (DWD). The

20 by means of atmospheric forcing fields from the ICON-EU model, provided by the German Weather Service (DWD). The horizontal resolution of ICON-EU is about 6.5 km and output is produced in 1-hour intervals.

#### **3 R500:** model results and validation

STPPs are generated in the straining field of mesoscale eddies. According to Osínski et al. (2010), the condition for eddy resolving models of the southern Baltic is that the Rossby radius is resolved by at least 4–5 horizontal nodes. As the Rossby radii in the Bornolm and Arkona Basins are in summer around 7.2 and 3.7 km, respectively (Fennel et al., 1991), and since the grid size of HBM is 1 nm, it is definitively not eddy-resolving or even eddy-permitting (2–3 nodes) in the Arkona Basin and perhaps eddy-permitting at best in the Bornholm Basin. The eddy-resolving R500 was initialised from HBM on 1 June 00:00 and integrated for 30 days, thus covering the entire survey period of the "Expedition Clockwork Ocean". The vertical mixing in R500 is accomplished with GLS using the *k* – ω setup of Wilcox (1988) with the stability function of Kantha and Clayson (1994).

Salinity is the ideal parameter to trace turbulent patterns because on the scales of atmospheric forcing (6.5 km) it depends solely on precipitation and evaporation. While the precipitation is constant within on grid cell of the atmospheric model, evaporation is impacted by the air-sea temperature difference, but the corresponding spatial variability of the salinity is negligible under realistic conditions; in particular, because a salinity increase due to evaporation is rapidly mixed in the vertical. By contrast, the sea surface temperature is controlled by the net surface heat flux and depends largely on the longwave radiation flux from the ocean, sensible and latent heat fluxes, which in turn are affected by the submesoscale variability of the threedimensional velocity field and vertical mixing. This may blur the sea surface temperature and, especially in June, superimpose

- a strong seasonal trend. Fig. 2 shows salinity in the uppermost layer of HBM and R500 on 1, 10, 20, and 30 June (left and centre panels). R500 rapidly develops turbulent structures that are only marginally identifiable in HBM. These are, for example, the low-salinity outbreaks along the northern boundary and a high-salinity eddy in the Arkona Basin, and mushroom-like patterns east and southeast of Bornholm on 10 and 20 June. The horizontal scales of these features are O(10 nm), but also smaller patterns with a horizontal extent of 5 nm, or even less, are generated by R500, such as the filaments around Bornholm and the
- meanders immediately off the Polish coast on 20 June. Hence, R500 apparently bridges the gap between the mesoscale and the submesoscale.

R500 provides the initial and boundary conditions for R100. Insofar, it is worth knowing to what extent its generated twodimensional turbulence is in a state of statistical equilibrium, and at what time during the integration an equilibrium state is attained. To determine this so-called spin-up time, the blue dash-dotted graph in Fig. 3 shows a time series of the domain-

- avageraged kinetic energy per unit mass,  $KE_{R500}$ ; it fluctuates strongly between  $3 \times 10^{-3}$  and more than  $12 \times 10^{-3}$  m<sup>2</sup> s<sup>-2</sup> and does not reach a stable value. Evidently, it is difficult to determine the spin-up time from R500 because  $KE_{R500}$  is strongly impacted by wind bursts as shown by the black curve. Therefore, R500 was compared to a run where the atmospheric forcing was completely turned off by setting the air-sea fluxes of net heat, fresh water and momentum to zero. This run is referred to as R500\_NF ("No Forcing"). Here, the intense fluctuations of KE vanished as shown by  $KE_{R500 NF}$  (blue solid graph), but there
- are still some smaller-scale oscillations with maxima on 11, 14, 20, and 28 June, the existence of which impede the estimate of a spin-up time. These oscillations are slightly correlated with the wind bursts lagging behind for about one day. Potentially, they are triggered by the remote forcing of HBM via the lateral boundaries which explains the time lag. Another cause could be vacillations of *KE* which is a well-known peculiarity of nonlinear rotating fluids (Hide, 1958; Früh, 2015). In order to filter out the oscillations, the cumulative average  $\overline{KE_{R500_NF}}$  (this is for all points in time the mean kinetic energy up to the actual
- point) was computed. This quantity attains a maximum of  $6 \times 10^{-3}$  on 1 June and then decreases to about  $3.4 \times 10^{-3}$  m<sup>2</sup>s<sup>-2</sup> on 7 June. Thereafter, it increases steadily to  $4.4 \times 10^{-3}$  on 22 June and remains constant until the end of the month. Hence, the growth phase of kinetic energy is about 15 days corresponding to an e-folding scale of 12 days that may be considered as the spin-up time.

The analysis of the prognostic fields of R500\_NF yielded an additional perception: the tracer fields exhibit much more spatial variability in comparison to the corresponding fields of R500 (see the right panel in Fig. 2). A rich variety of smaller details is revealed that were hidden in the forced run, such as the STPPs along the German and Polish coast, multiple filaments in the entire model domain, and sharper frontal gradients. On longer time scales, the larger scale patterns also develop differently. Similar findings were presented by the observational study of Kubryakov and Stanichny (2015): from an analysis of satellite altimeter data, they demonstrated that a weakening of the large-scale circulation leads to baroclinic instability of the Rim

35 Current in the Black Sea. An explanation for this behaviour was provided by the earlier studies of Zatsepin et al. (2005).

Literally, their laboratory experiments showed that "The development of an instability, the generation of meanders, and the formation of eddies are prevented by the Ekman divergent circulation caused by the wind, which presses the frontal current to the coast throughout the periphery of the tank, thus contributing to the significant increase in kinetic and available potential energy. When the wind forcing stops, the instability grows rapidly." While the above mentioned papers focused on mesoscale

instabilites, Renault et al. (2018) showed that the evolution of submesoscale instabilities in the Californian Upwelling System is damped by the wind stress as well. Another reason for the decrease of submesoscale activity in response to the increase of the wind speed was provided by Mahadevan et al. (2010), who showed that the wind field blurs the small-scale features by horizontal mixing. These findings are important because they guide the setup of the R100 experiments below.

A comparison with observations is shown in Fig. 4 by means of the surface salinity from R500 and a RGB composite derived

- from a satellite-borne multi-spectral image in the visible spectral range. In the latter, the patterns arise due to the presence of naturally occurring optically active substances like chlorophyll, which act like flow tracers. The image reveals three spiral-like patterns that may be considered as proxies for cyclonic (C1, C2) and anticyclonic (AC) eddies. The centre of C1 is almost at exactly the same location [15°30' E, 54°40' N] in the satellite image and in R500. The centre of C2 is at about [14°10' E, 55°7' N] in R500. Unfortunately, the central part of this cyclone is masked by clouds in the satellite image, but the unmasked
- portions provide evidence for its real existence. Strong peripheral fronts are visible both in R500 and the satellite images. The positions of AC differ the most: in Fig. 4b, the centre of that anticylone is at [15°50' E, 54°57' N], in Fig. 4a at about [15°40' E, 55°0' N], hence roughly 7 nm to the west. It is legitimate to associate these positions with each other because both in R500 and in the satellite snapshot, as AC is the anticyclonic "partner" of C1, both form a mushroom-like eddy pair. On the whole, the apparent similarities between the observations and the R500 simulations make the model output trustworthy. Moreover, the
- above mentioned eddies and fronts are already visible on 20 June in R500 (see Fig. 2) but not at all in HBM at the same time. Hence, in R500 the mesoscale environment is apparently better reproduced than in HBM.

#### 4 R100: results

Apparently, R500 is able to reproduce STPPs but only those in the low-wavenumber part of the spectrum. To provide more insight into the higher-wavenumber parts and to cover the entire period of the "Expedition Clockwork Ocean", R100 was
initialised from R500 on 15 June 00:00 and integrated until 29 June 00:00. Based on the experience with R500, the surface fluxes in R100 were turned off. This is indeed a strong simplification and the model results may differ significantly from the reality. However, it helps to understand the evolution and the dynamics of submesoscale processes, which are controlled primarily by potential vorticity conservation in the adiabatic limit (McWilliams, 2008). Different formulations and coefficients for the horizontal eddy viscosity and diffusivity were tested. The best results were obtained from a model run with mono-

30 harmonic mixing for the momentum and biharmonic mixing for the tracers (Table 2). This choice caused a minimum damping of STPPs, and a better representation of small-scale details without numerical noise. Moreover, as R100 is rather expensive in terms of computer resources, the vertical mixing was parameterised with KPP instead of the time-consuming GLS scheme. From a *KE* analysis, the spin-up time was an estimated 2 days. Hence, a 6-day spin-up period starting on 15 June was sufficient to produce statistically equilibrated fields at the beginning of the survey on 20 June.

In the following, the results of R100 are illustrated by Figs. 5 - 17, and 18b, c. All figures are snapshots taken at the specified day at 00:00.

## 5 4.1 Tracer patterns

patches in the southern region of the R100 domain.

The top-layer patterns of salinity and potential temperature are shown in the left and centre panels of Fig. 5 in 3-day intervals for the observational period. The top layer ranges between about 0.3 and 1.7 m. To asses the effect of the downscaling,the salinity patterns are compared with the corresponding ones of R500\_NF on 20 and 30 June (Fig. 2). The large-scale features resemble each other to a high degree, such as the low salinity to the west and the east of Bornholm and the higher-salinity pool south of the island. However, R100 exhibits much more variability in the submesoscale waveband and STPPs down to 1-km

south of the island. However, R100 exhibits much more variability in the submesoscale waveband and STPPs down to 1-km scales are resolved. The signatures of such small-scale features are better visible in the temperature patterns, such as the cold

The knowledge of the temporal change of the density (or buoyancy) gradient is essential for frontogenetic and frontolytic processes and filamentogenesis, that is expressed by the frontal tendency equation (Holton, 1982; Capet et al., 2008b; Gula et al., 2014). As a first approach, the absolute horizontal gradient of potential density,  $\rho$ ,

$$|\nabla\rho| = \sqrt{\left(\frac{\partial\rho}{\partial x}\right)^2 + \left(\frac{\partial\rho}{\partial y}\right)^2} \tag{1}$$

is calculated, where x and y are the Cartesian coordinates pointing to the east and to the north, respectively. This quantity is shown in the right column of Fig. 5. Narrow filaments indicate locations of intensified gradients, i.e. density fronts. From the sequence of the images, it appears that the quantity of the fronts increases with time and that the gradients amplify.
This is partly confirmed by a frequency distribution of |∇ρ|, shown in Fig. 6a: from 15 to 25 June, the frequency grows consistently in all frequency intervals. From 25 to 29 June, however, this behaviour changes. While the frequency distribution for |∇ρ| < 1.2 × 10<sup>-3</sup> kg m<sup>-4</sup> remains almost the same, it decays at the same time for larger values of |∇ρ|. Hence, there are less locations with strong gradients indicating a weakening of strong density fronts. In order to determine whether there is an upper limit for the growth of |∇ρ|, R100 was repeated but now the initial and open boundary conditions were taken from

- R500\_NF. The resulting frequency distribution (Fig. 6b) exhibits more occurrences of strong gradients than in panel a on 15 and 20 June. This is plausible, because the atmospheric forcing in R500 has blurred the gradients. The frequency distribution of 25 June is almost identical to that of 20 June, and from 25 to 29 June it decays in the same way as in panel a. Hence, a "frontal arrest" occurs apparently when the strongest gradients approach a critical value around  $3 \times 10^{-3}$  kg m<sup>-4</sup> that is reached on 25 June in panel a and already on 20 June in panel b. However, it is not clear whether physical processes or numerical diffusion
- (or both) limit the increase of gradients. For the physical processes,  $\mathbf{Q}_w$ , the straining deformation by the vertical velocity (see eq. (7) below), would be a suitable candidate, because downwelling on the dense side of a front and upwelling on the less dense side lead to a weakening of the cross-front density contrast. Another process was identified by Sullivan and McWilliams (2018) and Gula et al. (2014), who investigated the life cycle of a dense filament. There, frontogenesis is arrested at a small

width of about 100 m. It is mostly driven by an enhancement of turbulence through submesoscale horizontal shear instabilities that draw their energy primarily from the horizontal mean shear via the horizontal Reynolds stresses.

#### 4.2 Kinematics

Fig. 7 shows the near-surface properties of the velocity field and derived quantities on 26 June. This day was chosen because it represents the conditions in the middle of the survey period and it allows comparisons with Fig. 5.

## 4.2.1 Horizontal currents

The horizontal velocity field (Fig. 7a) is separated into two regimes: high current speeds frequently exceeding 0.1 m s<sup>-1</sup> are found north of about 54°50' N, south of 54°35' N, and west of 14°40' E. They are organised in jet-like streaks or circular features, and their positions are clearly related to the main density fronts shown in Fig. 5. Maximum speeds of 0.32 m s<sup>-1</sup> are

10 found in the anticyclonic eddy southeast of Bornholm. The high-speed regime surrounds a pool of water with speeds lower than  $0.1 \text{ m s}^{-1}$  in the centre of the model domain. The transition between the high-speed and low-speed regimes appears to be related to the 40-m depth contour (the white line in Fig. 7a), at least in the south and west. Potentially, the bathymetry is acting as a waveguide for the westward jet south of Bornholm. One may note that both the anticyclone AC and the cyclonic eddy C1 are at about the same position as in Fig. 4a three days before.

#### 15 4.2.2 Relative vorticity

Information about the dynamical properties of the flow field is derived from the relative vorticity  $\zeta$ , i.e. the vertical component of the curl of the total velocity:

$$\zeta = \left(\nabla \times \mathbf{V}\right)_z = \frac{\partial v}{\partial x} - \frac{\partial u}{\partial y} \quad . \tag{2}$$

Here, z is the vertical coordinate and u, v, w are the zonal, meridional, and vertical components of the total velocity vector V.

- Fig. 7b reveals the whole range of relative vorticity (scaled by f) between the largest scales of about 20 km and the smallest scales on the order of the grid size of 100 m. The dominant mesoscale patterns are the large anticyclonic relative vorticity patch southeast of Bornholm and the wavelike structures to the south that belong to the westward jet mentioned above. Further west and south in the shallow water, pools of concentrated cyclonic relative vorticity vary mostly between 5 and <1 km in size. The smallest visible features are cyclonic streamers, with width approaching the Nyquist scale of 200 m.
- There is a strong asymmetry between the distributions of cyclonic and anticyclonic relative vorticity. The cyclonic relative vorticity is organised in thin filaments, comma-like "hooks", and quasi-circular pools of water. In contrast, the distribution of the anticyclonic relative vorticity is smoother, and the occupied area is larger than the area utilised by  $\zeta/f > 0$ , which is in agreement with McWilliams (2016). The frequency distribution shows that  $\zeta/f < 0$  in 63 % of the grid cells, while positive values are found only in 37 %. Moreover, in less than 0.2 % of the grid cells,  $\zeta/f < -1$ , indicating negative potential vorticity
- 30 (Dewar et al., 2015; Gula et al., 2016b) and associated inertial instability.

The relative vorticity structures north of 55° N exhibit lamellar patterns that are probably caused by false advection from the parent domain, although cubic splines were applied to all prognostic variables in the context of the downscaling. Such patterns are only visible in the relative vorticity patterns, while the horizontal velocities (Fig. 7a), and the tracer fields (Fig. 5) look rather reliable. Obviously, the downscaling did a good job with the prognostic variables, but along the open boundaries it may not have been able to reflect the first derivatives correctly.

#### 4.2.3 Divergence and vertical motion, impact of bathymetry

The horizontal divergence of the velocity,

$$\delta = \left(\nabla \cdot \mathbf{V}\right)_h = \frac{\partial u}{\partial x} + \frac{\partial v}{\partial y} \quad , \tag{3}$$

is displayed in Fig. 7c. It can be divided into 3 different classes of textures: Class I structures are thin, less than 1-km wide

- filaments of negative divergence (convergent flow) that are bordered by wider areas of divergent flow on one or both sides. The major density fronts (Fig. 9) are aligned with such filaments. On the contrary, the areas of convergent flow that are not congruent with enhanced horizontal density gradients belong to Class II. In particular, those patterns tagged by the yellow arrows in Fig. 9 are not correlated with enhanced density gradients (note that the value of the magenta contour refers already to rather low gradients, cf. Fig. 5). It is conjectured that these features are related to internal waves generated during the genesis
- of nearby fronts and jets (Plougonven and Snyder, 2007; Shakespeare and Taylor, 2014; Shakespeare, 2019). This is confirmed by Fig. 10: the vertical displacements of isopcnals between  $15^{\circ}07'$  E and  $15^{\circ}09'$  E are clear indicators for internal waves with a wavelength around 1000 m. In order to verify that these patterns are caused by internal waves, their frequency at 10-m depth was estimated to about  $4 \times 10^{-4}$  s<sup>-1</sup>, somewhat less than the buoyancy frequency at the same depth ( $\approx 5 \times 10^{-4}$  s<sup>-1</sup>) but larger than the Coriolis frequency. Hence, the patterns are indeed intermediate-frequency internal waves. However, a closer
- inspection is not advisable, because ROMS is hydrostatic and various properties of internal waves are correctly reproduced only in nonhydrostatic models (Wadzuk and Hodges, 2009; Vitousek and Fringer, 2011).

Class III textures are small patches of either sign and horizontal scales of O(1) km. They are found almost everywhere, but their highest values are found in the area of higher current speeds in the northwest corner of the domain and in the anticyclonic eddy southeast of Bornholm. The topography of potential density surfaces in the anticyclone shows that the notables are accompanied by large expansions of iconvensels, indicating interest interest interest interest.

patches are accompanied by large excursions of isopycnals, indicating intense internal wave activity. These waves are emitted from fronts at the periphery of the eddy, and the superposition of internal waves coming from different directions leads to the rugged shape of isopycnals and contemporaneously to the patchy pattern that is in agreement with the findings of Väli et al. (2018).

In ROMS, the vertical speed is computed from the continuity equation. As expected, the spatial patterns of the vertical speed 30 are therefore identical with that of the divergence, but only for the divergence in the top layer and the vertical speed at the base of that layer. In contrast, the horizontal distribution of the vertical speed at 5-m depth (Fig. 7d) still resembles the top layer divergence, but with significant differences. The most striking discrepancy is that the thin filaments of convergent flow belonging to Class I (Fig. 7c) are not at all reflected by the vertical motion pattern. On the other hand, the vertical motions related to the internal wave packets discussed above are still clearly visible. To explain this disagreement, Fig. 10 shows a zonal section of the vertical speed, together with the potential density and the meridional velocity. The vertical motion induced by the internal waves attains speeds of up to  $\pm 20$  m day<sup>-1</sup> and the signal is pronounced over the entire vertical range between the surface and 10-m depth; by contrast, the magnitude at the location of the front barely exceeds 5 m day<sup>-1</sup> and the vertical

- speed diminishes with increasing depth. Apparently, the major surplus (deficit) of mass created by the frontal convergence (divergence) at the surface, is discharged (imported) by the frontal jet or the secondary divergent circulation (cf. Fig. 1 in McWilliams et al. (2009), or Bleck et al. (1988)). As such a horizontal discharge/import mechanism is not available in a pure internal wave field, the only way to respond to any non-zero divergences is by vertical motion; this is explained in more detail in Section 4.3.4.
- In Section 4.2.1, it was already mentioned that the horizontal current patterns are somehow correlated with the water depth. Such a relationship applies as well for the patterns of the other variables displayed in Fig. 7. The spatial variability of relative vorticity and the frequency of narrow filaments of confluent flow are clearly enhanced in the shallow water regions. In contrast, the texture of the vertical speed is rather smooth there, indicating less internal wave activity.

#### 4.3 Impact of atmospheric forcing

In order to justify the approach to run R100 without atmospheric forcing, R100 was repeated but now including all fluxes at the air-sea interface as provided by the ICON-EU model. The results for 26 June of the same parameters as in Fig. 7 are displayed in Fig. 8.

The patterns of the horizontal velocity (Figs. 7a and Fig. 8a) resemble each other but only in the "high-velocity regimes" west and east of Bornholm. Here, the direction of the currents did not change significantly but the maximum speeds increased from

- 0.32 to about 0.42 ms<sup>-1</sup>. In contrast, the pattern in the low-speed pool in the centre of the model domain changed dramatically. While there in the no-forcing run the maximum speeds rarely exceeded 0.1 ms<sup>-1</sup>, the highest speeds are now about twice as high. Moreover, the wind stress caused a unidirectional flow to the east as opposed to alternating flow directions in Fig. 7a. The atmospheric forcing impacts severely the top-layer relative vorticity patterns (panels b of the respective figures). Submesoscale features no more detectable, i.e. there are no filaments, comma-like "hooks", and circular pools of cyclonic vorticity. However,
- some mesoscale structures are still visible, for instance the large anticyclonic eddy southeast of Bornholm and the small cyclone close to the eastern boundary at about 54°45' N. Overall, the atmospheric forcing acts like a filter that wiped out any structures with scales of less than about 10 km. This smoothing effect also removed the Class I divergence patterns (panels c). In contrast, the number of small-scale patches (Class III) and their amplitude has increased. This is also reflected by the vertical velocity (panels d). Presumably, the imposed momentum stress enhanced the internal wave activity.
- In summary, not before the atmospheric forcing is switched off, STTPs start to grow. This reflects a situation which also occurs in Nature when the wind subsides. For instance, applying a high-resolution Princeton Ocean model to the Gulf of Finland, Zhurbas et al. (2008) and Väli et al. (2017) demonstrated that the horizontal eddy viscosity increased and submesoscale cyclonic vortices evolved as soon as the wind slackened.

#### 4.4 Fronts and eddies

#### 4.4.1 Frontogenesis and eddy formation

The right column of Fig. 5 shows snapshots of  $|\nabla \rho|$ . A statistical analysis indicates an increase of the frequency of occurrence of high density gradients and a general sharpening of the density fronts with time. However, the latter provides only a statistical

information and it does not show the locations were the fronts are sharpening (frontogenesis) or weakening (frontolysis) and which physical processes are contributing thereto. The missing information is conveyed by the frontal tendency equation

$$F = \frac{d}{dt} |\nabla \rho|^2 \quad , \tag{4}$$

which was introduced by Hoskins (1982) into the meteorological literature. F describes whether the absolute horizontal density gradient of a Lagrangian water parcel is growing (F > 0) or weakening (F < 0) with time t. Using the nomenclature of Capet et al. (2008b), F is further decomposed as

$$F = (\mathbf{Q}_h + \mathbf{Q}_w + \mathbf{Q}_{dv} + \mathbf{Q}_{dh}) \cdot \nabla \rho \qquad .$$
<sup>(5)</sup>

Here, the terms are defined as

$$\mathbf{Q}_{h} = -\left(\frac{\partial u}{\partial x}\frac{\partial \rho}{\partial x} + \frac{\partial v}{\partial x}\frac{\partial \rho}{\partial y}, \frac{\partial u}{\partial y}\frac{\partial \rho}{\partial x} + \frac{\partial v}{\partial y}\frac{\partial \rho}{\partial y}\right),\tag{6}$$

which is the vector representing the straining deformation by the horizontal total velocity. The vector may be separated into 15 contributions  $\mathbf{Q}_{geo}$  from the geostrophic and  $\mathbf{Q}_{ageo}$  from the ageostrophic velocity, respectively.

$$\mathbf{Q}_{w} = -\frac{\partial \rho}{\partial z} \left( \frac{\partial w}{\partial x}, \frac{\partial w}{\partial y} \right) \tag{7}$$

is the equivalent quantity for the vertical velocity, and

$$\mathbf{Q}_{dv} = \left(\frac{\partial\rho}{\partial z}\frac{\partial^2 A_V^{\rho}}{\partial x \partial z} + \frac{\partial A_V^{\rho}}{\partial z}\frac{\partial^2 \rho}{\partial x \partial z}, \frac{\partial\rho}{\partial z}\frac{\partial^2 A_V^{\rho}}{\partial y \partial z} + \frac{\partial A_V^{\rho}}{\partial z}\frac{\partial^2 \rho}{\partial y \partial z}\right)$$
(8)

is the contribution from vertical mixing where  $A_V^{\rho} = A_V^{\rho}(x, y, z, t)$  is the eddy diffusion coefficient for density. Finally,

$$\mathbf{Q}_{dh} = \left(\frac{\partial D}{\partial x}, \frac{\partial D}{\partial y}\right)$$
 (9)

is the effect of the horizontal diffusion of density, D.

The components of the tendency equation are displayed in Fig. 11 for a subregion of the R100 domain. The quantities F, Q<sub>geo</sub>, Q<sub>ageo</sub>, Q<sub>w</sub>, Q<sub>dv</sub>, and Q<sub>dh</sub> at 2-m depth are shown in the centre and bottom rows. They are the scalar products of the corresponding vectorial quantities Q and ∇ρ. For comparison with the dynamical background, the top-layer patterns of |∇ρ|,
ζ/f, and the vertical velocity w at the base of the top layer are displayed in the top row of the same figure. The subplots of |∇ρ| and ζ/f reveal four major frontal systems, F1–F4, and two cyclonic vortices, C3 and C4. The width of all fronts and the radius of the circular cyclone C3 (defined by the zero-crossings of ζ/f) is ≤ 2 km, while the semi-axes of the elliptical-shaped

cyclone C4 vary between 2 and about 4 km. Both frontogenetic and frontolytic processes are roughly evenly distributed and of the same order of magnitude for F,  $Q_{geo}$ , and  $Q_w$ . This is different for  $Q_{ageo}$  and  $Q_{dv}$  which are primarily frontogenetic. Moreover,  $Q_{dv}$  only differs significantly from zero at the fronts, while at the positions of C3 and C4, it is close to zero. By contrast, the contribution of  $Q_{dh}$  to the tendency F is everywhere negligible. F exhibits a bimodal pattern for F1 and F2, with

- 5 F < 0 on the anticyclonically sheared side and F > 0 on the cyclonic side. A comparison of the patterns of the components of F reveals that F > 0 in F1 and in the western and central parts of F2 is predominantly supported by  $Q_{ageo}$  and  $Q_{dv}$  and to a lesser extent by  $Q_{geo}$ , while the sign of  $Q_w$  is alternating along-front. On the other hand, F 
- 15 these sites that the coast is on the right, relative to the current direction, and that the relative vorticity is anticyclonic due to the no-slip condition at the lateral solid boundaries. However, at sites no. 2 and 5, the curvature of the coastline is anticyclonic in contrast to the other sites. Apparently, a solid boundary on the right and anticyclonic curvature of that boundary, are are additional necessary conditions for the formation of eddies driven by inertial instability. These conditions are also satisfied at site no. 7, but there is no indication for eddy growth. It is most likely prevented by the southward current with cyclonic vorticity
- along the west coast of the island. The above findings are in agreement with Väli et al. (2017) who found values of ζ/f < −1 at various near-coast locations in the Gulf of Finland. Gula et al. (2016b) showed that equivalent conditions were satisfied in the Gulf Stream where the anticyclonic shear is amplified by the topographic drag against the slopes of the Great and Little Bahama Banks on its way through the Florida Straits. A similar situation is given where the California Undercurrent passes along Point Sur Ridge, a topographic obstacle near Monterey Bay (Dewar et al., 2015; Molemaker et al., 2015). Both cases</li>
  lead to the formation of unstable submesoscale fronts and eddies.

According to Thomas et al. (2013), symmetric instability occurs when

$$\phi_{Ri_{qeo}} < \phi_c \quad , \tag{10}$$

with

$$\phi_{Ri_{geo}} = \tan^{-1} \left( -\frac{1}{Ri_{geo}} \right) \tag{11}$$

30 and

$$\phi_c = \tan^{-1} \left( -\frac{\zeta_{geo}^a}{f} \right) \quad . \tag{12}$$

Here,

$$Ri_{geo} = \frac{f^2 N^2}{|\nabla_h b|^2} \tag{13}$$

is the Richardson number of the geostrophic flow,

$$b = -\frac{g\rho}{\rho_0} \tag{14}$$

5 is the buoyancy,

$$N^2 = -\frac{g}{\rho_0} \frac{\partial \rho}{\partial z} \tag{15}$$

the squared Brunt-Väisälä frequency, and

$$\zeta_{geo}^a = f + \frac{\partial v_{geo}}{\partial x} - \frac{\partial u_{geo}}{\partial y} \tag{16}$$

the absolute vorticity of the geostrophic flow  $V_{geo} = (u_{geo}, v_{geo})$ .  $\rho_0$  is a constant reference density, and g the gravitational 10 acceleration. Specifically, in regions where  $\zeta_{qeo}^a/f > 1$  (cyclonic vorticity), symmetric instability prevails, if the conditions

$$C_{SI} = -90^{\circ} < \phi_{Ri_{geo}} < \phi_c \qquad \land \qquad \phi_c < -45^{\circ} \tag{17}$$

are satisfied. By contrast, symmetric instability is the dominant mode of instability in regions of anticyclonic vorticity ( $\zeta_{geo}^a/f < 1$ ), if

$$C_{SI} = -90^{\circ} < \phi_{Ri_{qeo}} < -45^{\circ} \qquad \wedge \qquad \phi_c > -45^{\circ} \qquad . \tag{18}$$

- As the condition described by eq. (10) must be satisfied in the mixed layer for symmetric instability to occur, all above quantities were evaluated in R100 at 3-m depth. Note that the relative vorticity is identical to the relative vorticity of the geostrophic flow, because ζ is the curl of the rotational, divergence-free part of the total velocity which is the geostrophic velocity by definition. Hence, a decomposition of the total velocity in a geostrophic part V<sub>geo</sub> and an ageostrophic part V<sub>ageo</sub> requiring a Poisson solver is redundant (for some test cases, both ζ<sub>geo</sub> and ζ were evaluated showing identical results). For simplification, the subscript "geo" will therefore be dropped for ζ<sup>a</sup><sub>geo</sub> and Ri<sub>geo</sub>. Fig. 14 shows φ<sub>Ri</sub>, φ<sub>c</sub>, and C<sub>SI</sub> at the start of the model integration on 15 June (using as initial condition when the corresponding field is mapped from R500 on the 100-m grid), and on 26 June. On 15 June, there are large areas where -90° < φ<sub>Ri</sub> < -45° (Fig. 14a, in blue), indicating satisfaction of the first condition of eqs. (17) and (18). However, this condition is only necessary and sufficient for regions of cyclonic vorticity (φ<sub>c</sub> < -45°). Considering the second condition in eq. (18), necessary and sufficient conditions for regions.</p>
- of anticyclonic vorticity are satisfied, if, and only if,  $\phi_c > -45^\circ$ . Linking the requirements for  $\phi_c$  (Fig. 14c) and  $\phi_{Ri}$  yields the logical map Fig. 14e for 15 June, indicating where eqs. (17) and (18) are satisfied for cyclonic and anticyclonic vorticity, respectively. Compared to the fairly widespread areas where the necessary conditions for  $\phi_{Ri}$  and  $\phi_c$  are met separately, the necessary and sufficient conditions for  $C_{SI}$  are satisfied only in rather limited streaks. Nine days later, on 26 June, there are

just a few spots where even  $C_{SI}$  is satisfied (Fig. 14f). Hence, favourable conditions for symmetric instability prevail only during the initial phase of the model integration, becoming less frequent thereafter. More insight into the temporal evolution of the statistics of  $C_{SI}$  is provided by the relative frequency of occurrence,  $p(C_{SI}) = 100 \cdot n(C_{SI} \equiv true)/N$ , where n is the number of grid cells where the criterion for  $C_{SI}$  is met, and N is the total number of grid cells.  $p(C_{SI})$  decays from 4.8 %

- on 15 June to 1.8 % on 16 June, which yields an e-folding scale of  $\approx$  1 day comparable to the inertial period of 14.6 h at this latitude. Thereafter,  $p(C_{SI})$  decreases almost steadily and stabilises around 0.5 %. Hence, as symmetric instability extracts kinetic energy from the geostrophic flow at a rate given by the geostrophic shear production (Thomas et al., 2013), this process comes rapidly due to a halt after about one inertial period and mixed-layer instability, presumably, takes over.  $p(C_{SI})$  was also computed for the R100 model run with atmospheric forcing (see above Section 4.3). In that run,  $p(C_{SI})$  increased strongly on
- 18, 21, 24, and 27 June, which were the times of the wind bursts (cf. Fig. 3), and slackened thereafter. Thus, the conditions favouring symmetric instability are re-established during strong wind events by the buildup of kinetic energy, and as soon as the wind slackens, the energy is released.

Mixed-layer instability is an efficient mechanism to restratify the mixed layer (Boccaletti et al., 2007). For the determination of the mixed-layer depth, MLD, a  $\Delta \sigma_{\theta} = 0.1$  kg m<sup>-3</sup>- criterion was used, i.e. MLD was defined as the depth where the potential

- density exceeds the surface density for the first time by  $\Delta \sigma_{\theta}$ . The domain-wide mean MLD decreases almost linearly from 12.7 m on 15 June to 5.1 m on 29 June. Hence, the mixed-layer is restratified, probably by mixed-layer instability. In order to exclude that symmetric instability drives the restratification, the relative frequency  $p(Ri) = 100 \cdot n(Ri < 1)/N$  was computed. This quantity drops from 23.2 to 4.0 % within the first day of the integration and decreases thereafter slowly to 1.4 % on 29 June. Hence, after about one inertial period,  $Ri \ge 1$  almost everywhere, which makes symmetric instability unlikely. This is
- in accordance with Haine and Marshall (1998) who state that "symmetric instability rapidly generates a layer with vanishing potential vorticity (Ri = 1), but non-zero vertical stratification. Thereafter a nonhydrostatic baroclinic instability develops ...".

#### 4.4.4 Frontal circulation

In order to provide details of the secondary circulation in submesoscale fronts in a non-idealised model, features resembling the confluence situation as provided by a deformation field were identified in the R100 fields of 26 June in the red box indicated in

Fig. 7d. Here, dense water in the east and light water in the west, form a confluent flow pattern and generate a density contrast at the surface of more than 0.4 kg m<sup>-3</sup> over a horizontal distance of about 2 km within the black rectangle shown in Fig. 15.

Detailed maps of various quantities within that rectangle are displayed in Fig. 16a–e, and vertical cross sections along the dashed lines are shown in Fig. 16f–j. In the western part of the front, the total velocity vectors at the surface are aligned almost parallel to the isopycnals (panel a); extreme values of  $|\mathbf{V}|$  are close to 12 cm s<sup>-1</sup> at the surface. In the cross section (panel

f), the *v* component attains minimum values of < -10 cm s<sup>-1</sup>, located by a few hundred metres to the west of the maximum horizontal density gradient (panel b). The velocity field depicts the classical picture of a frontal jet, with a width of about 3 km and the depth around 4 m. It is defined by the -5 cm s<sup>-1</sup> isotach of the *v*-component. The horizontal shear of the jet creates a front-parallel band of strong cyclonic relative vorticity with maximum values of > 1.6*f* (panels c, h). The cross-front width of the band is about 600 m, and the highest values of the relative vorticity are congruent with the maximum horizontal

density gradient (panel b). A wide band of anticyclonic relative vorticity extends to the west of the cyclonic region. East of the cyclonic region, several bands with alternating sign are aligned. Vectors of  $\mathbf{V}_{geo}$  are superimposed in panel c. A comparison with panels a and c reveals, that the directions of  $\mathbf{V}$  and  $\mathbf{V}_{geo}$  resemble closely each other in appearence, but the speeds are slightly different. While the maximum speeds of  $\mathbf{V}$  are around 12 cm s<sup>-1</sup> in the jet, those of  $\mathbf{V}_{aeo}$  are close to 10 cm s<sup>-1</sup> i.e.

- 5 the jet is super-geostrophic (Persson, 2001). This is confirmed by panel d which shows the horizontal divergence  $\delta/f$ , together with the vectors of the ageostrophic flow  $\mathbf{V}_{ageo}$ : the meridional components of  $\mathbf{V}_{ageo}$  and  $\mathbf{V}_{geo}$  have the same (negative) sign, hence  $\mathbf{V}_{ageo}$  amplifies the geostrophic jet. The maximum ageostrophic speeds are around 6 cm s<sup>-1</sup>. In the cross-frontal direction (panel a), the 5.25 kg m<sup>-3</sup> isopycnal separates two regimes of opposite zonal component of the ageostrophic flow,  $u_{ageo}$ : in the lighter water,  $u_{ageo} > 0$ , and  $u_{ageo} < 0$  in the heavier water. While the overall magnitude of  $u_{ageo}$  ist just a few
- 10 mm s<sup>-1</sup>, it attains its highest value of about 1 cm s<sup>-1</sup> right at the location of the surface outcrop of the 5.25 kg m<sup>-3</sup>-isopycnal.

The impact of  $V_{ageo}$  on the divergence  $\delta$  and on the vertical speed w is illustrated in Figs. 16d, i, e, and j. According to panels d and i, extreme values of  $\delta/f \approx 0.3$  and  $\delta/f \approx -0.8$  are found at the sea surface immediately to the west and to the east of the maximum horizontal density gradient, respectively. The minimum of the divergence (=maximal convergence) drives a deep-reaching downwelling cell with extreme speeds of almost 15 m day<sup>-1</sup>. Less intense upwelling of maximal  $\approx 6$  m day<sup>-1</sup>

- 15 is associated with the divergence maximum (panel j). The positions of the extrema of δ/f are also reflected by its horizontal surface distribution in panel d, while a comparison of panels d and e reveals a clear correlation between δ and w. Patches of convergent motion are found also along the 5.25 kg m<sup>-3</sup> isopycnal in panel i; potentially, they feed the large downwelling area underneath it. Further cells of intense vertical motion are found west and east of the surface front as shown in panel j. Apparantly, these patterns are not parts of the secondary circulation, but rather caused by internal waves (see Section 4.2.3) or
  20 by another weak front located about 600 m further east
- 20 by another weak front located about 600 m further east.

Overall, the described secondary circulation pattern closely resembles the ones of the idealised model studies mentioned in the Introduction; primarily, this is a "single overturning cell with upwelling and surface divergence on the light side and downwelling and surface convergence on the dense side" (literally after McWilliams (2016)). However, Fig. 16 provides additional details which were potentially not yet highlighted before:

- The frontal jet is super-geostrophic, i.e. the jet speed is amplified by an ageostrophic component. Under the assumption that the jet was in geostrophic balance, the theoretical inclination of the front was calculated as 0.003° using the Margules equation (Margules, 1906), the geostrophic jet speed, and the cross-front density difference. Assuming the σ<sub>θ</sub> = 5.25 kg m<sup>-3</sup>-isopycnal as the frontal interface, the tilt angle of that isopycnal was, however, only 0.001°. Hence, the slope of the front is not in geostrophic balance with the jet. This circumstance was not considered in the previous literature, assuming two-dimensionality (Bleck et al., 1988), or quasi-geostrophic (Nagai et al., 2006), or semi-geostrophic (Thompson, 2000) balance. In contrast, in the model of McWilliams (2017) (see Fig. 5 there), the ageostrophic contribution to the jet speed amounts to about 3 cm s<sup>-1</sup> while the geostrophic fraction is close to 12 cm s<sup>-1</sup>.
  - The maximum speed of the cross-frontal ageostrophic velocity is weaker than expected from earlier models. Right at the surface, it is only 1 cm s<sup>-1</sup>, while Bleck et al. (1988) arrived at > 4.5 cm s<sup>-1</sup> and McWilliams (2017) at least at 2 cm

 $s^{-1}$ . A potential reason for this discrepancy is that the other studies did not consider the ageostrophic velocity as being the ageostrophic part of the total flow, but rather as the departure from the (barotropic) deformation velocity or from the far-field average, instead. In the present situation, it was not possible to separate the total flow into such components because they were not defined. Hence, one may speculate that something like the deformation velocity is opposed to the cross-front ageostrophic velocity and attenuates it.

- 5
- According to Fig. 16g, the cross-frontal velocity is positive (eastward) in the light water ( $\sigma_{\theta} < 5.25 \text{ kg m}^{-3}$ ) and westward in the denser water below. Thus, the velocity converges at that isopycnal, in accordance with Fig. 16i. It is not known to the authors whether such a "sloping convergence" was mentioned before in the oceanographic literature.

The investigated front satisfies the criteria to be denoted as "submesoscale". The first criterion,  $Ro \sim O(1)$ , is confirmed by 10 Figs. 16c and h, indicating that the *f*-scaled relative vorticity (which is equivalent to a local Rossby number) is O(1). Another criterion is  $Ri \sim O(1)$  (Thomas et al., 2008; Mahadevan, 2016). According to Fig. 17,  $Ri \approx 2$  in the frontal region at the location of maximum convergence at the sea surface.

#### 5 Comparison of features with observations

problematic.

The R100 results presented above have provided a detailed insight into STPPs, such as tracer patterns, kinematic structures, and
dynamical processes related to fronts and eddies. A comparison with obervations, however, is rather limited because STPPs are difficult to measure due to their small spatial and temporal scales.

Kinematic quantities of STPPS were obtained from direct measurements in the framework of the LatMix and SubEx experiments. Shcherbina et al. (2013) presented a detailed view of submesoscale vorticity, divergence, and strain statistics from synchronous two-ship ADCP (Acoustic Doppler Current Profiler) samplings. Specifically, their observations indicated flows 20 with  $Ro \sim O(1)$  and an asymmetry in the distribution of the relative vorticity skewed towards positive values. The latter is in excellent agreement with the R100 results. By contrast, Ohlmann et al. (2017) identified STTPS with aerial guidance and seeded them with drifters. The Lagrangian observations exhibited high values of relative vorticity and divergence exceeding 5f, suggesting vertical velocities up to 240 m day<sup>-1</sup>. Such values are rather close to the values obtained from R100. Similar values for  $\zeta/f$  and  $\delta/f$  resulted also from the high-frequency radar observations of Parks et al. (2009). As the observations 25 mentioned above are two-dimensional and confined to the sea surface, they provide only limited information of subsurface properties of STPPs. Somewhat more detailed insight is gained from the high-resolution in-situ measurements of Zhong et al. (2017) which exhibit submesoscale kinematic structures along a vertical section through a 200 km wide mesoscale eddy in the South China Sea. Unfortunately, that eddy is an anticyclone and a comparison with the corresponding quantities in C3 is

30 The formation of a the submesoscale cyclonic eddy C3 (Fig. 12) resembles closely the sequence of infrared images published by Munk et al. (2000) (see Fig. 12 there) and the snapshot of Buckingham et al. (2017). In both cases, the eddies originate from an unstable thermal front which finally breaks up in a train of cyclonic eddies. While the diameters of the eddies shown by Munk et al. (2000) are around 10 km (very rough visual estimate), those of Buckingham et al. (2017) are smaller (1–10 km according to the authors).

A closeup infrared image of an eddy observed in the Southern California Bight is shown in Fig. 18, together with corresponding properties from the modelled cyclone C3 (see Figs. 11, 12). The observed sea surface temperature (panel a) exhibits
a spiraliform cyclonic eddy with a diameter of about 1 km, where the diameter is estimated as the width between the ≈ 13.8 °C isotherms (purple) on either side. Special features are the low temperature patches (black) close to the eddy centre suggesting upwelling, and the bright ripples in the southwest which are probably caused by internal waves. One may note also the cold spots "S" along the periphery; they are definitely not caused by a malfunction of the camera, because they show up at different

positions in a sequence of images taken within a period of about 20 minutes. Potentially, they are created by intense vertical

- 10 mixing (cf. Figure 2 in Marmorino et al. (2018)). For comparison with C3, the potential temperature and salinity at 15 m depth are provided in panels b and c. This depth was selected as the mixed-layer depth at the corresponding position in R100 is about 9 m and the surface signal of C3 is hardly resolved (cf. potential temperature and salinity on 26 June in Fig. 5). The overall structures of the C3 salinity and the sea surface temperature of the observed eddy are looking similar but the smaller details visible in the latter are not reproduced by C3: these are the ripples in the southwest, the cold spots, and as well the texture of
- 15 the tracer field in the core. Most probably, this is due to the insufficient horizontal resolution of R100 or non-hydrostatic effects which are not included in the applied ROMS version.

There are two observational studies available regarding necessary instability criteria at fronts (Thomas et al., 2013; Zhong et al., 2017) but they are not comparable to the maps in Fig. 14 because the criteria were computed along vertical sections across the Gulf Stream front and in a large anticyclonic eddy in the South China Sea, respectively. Concerning frontogenetic or frontolytic processes, computations or estimations of the components of the frontal tendency equation from observational data are not known to the authors.

Comprehensive observations of the frontal circulation (Pollard and Regier, 1992; Rudnick, 1996; Pallàs-Sanz et al., 2010) confirm the secondary circulation pattern as predicted from theoretical considerations and numerical models. However, the above studies did not resolve STTPs and the extrema of the vertical velocity are correspondingly low with  $O(10 \text{ m day}^{-1})$ .

- Namely, the vertical velocities in the front shown in Fig. 16e and j are on the same order but this is due to the low water depth. On the contrary, the higher-resolution observations of a submesoscale front oriented along the periphery of a mesoscale eddy (Adams et al., 2017) confirm vertical velocities of  $\mathcal{O}(100 \text{ m day}^{-1})$ . Moreover, it was demonstrated that within the same front existed confluent and diffluent regions of the cross-frontal velocity. This does not necessarily prove that the ageostrophic velocity changes sign, but it indicates that the generally accepted picture of the secondary circulation is only valid in the case
- 30 of frontogenesis driven by an externally imposed deformation field.

### 6 Conclusions

20

A double one-way nesting approach is used in order to simulate submesoscale turbulent patterns and processes (STPPs) in the southern Baltic Sea in summer 2016. In order to reproduce the mesocale environment in a realistic way, the Regional Ocean Modeling System (ROMS) with 500-m horizontal resolution (R500) is nested in an existing operational model, and further downscaling to a grid size of 100 m (R100) enables the generation of STPPs. In R500, the kinematic and dynamical structures are rather sensitive to the surface boundary conditions. While the response of mesoscale patterns to the turning-off of the atmospheric forcing is rather sluggish, it has an immediate impact on the generation of smaller-scale features which

5 represent already the low-wavenumber part of submesoscale turbulence in the spectral range around 5 km. Consequently, as R100 is intended to enable the generation of STPPs and to analyse their kinematic and dynamical properties und quasi-adiabatic conditions, the atmospheric forcing is turned-off from the outset.

After initialisation, the horizontal density gradients in R100 grow for about 10 days, and afterwards the frequency of strong gradients begins to decline, indicating frontal arrest as soon as the absolute horizontal density gradient reaches a critical value.

10 STPPs develop rapidly within about a day; they are characterised by relative vorticities and divergences reaching multiple of the Coriolis parameter, and strong vertical speeds of  $\mathcal{O}(100)$  m day<sup>-1</sup>. The frequency distribution of relative vorticity is clearly biased towards negative (anticyclonic) values. The horizontal divergence and the vertical motion are predominantly controlled by the velocity field associated with internal waves and to a lesser extent by frontal secondary circulations.

Typical elements of the secondary circulation of two-dimensional strain-induced frontogenesis are identified at an exemplary front in shallow water; these are the frontal jet, the downwelling on the dense side and upwelling on the less dense side. In addition to the results of idealised two-dimensional models, details of the ageostrophic current field and the related divergence are revealed: the frontal jet is not in geostrophic balance but super-geostrophic, instead. Further on, it is shown that a region of enhanced convergent flow is aligned with the slope of the frontal surface.

The components of the tendency equation are evaluated in a subregion of the R100 domain. At fronts, frontogenetic and 20 frontolytic processes represented by the frontogenetic tendency, F, and the contributions from the straining deformation of the 20 geostrophic and vertical velocity,  $Q_{geo}$  and  $Q_w$ , respectively, are equipartioned, bimodal, and of the same order of magnitude. By contrast, the contributions from the ageostrophic velocity,  $Q_{ageo}$ , and from vertical mixing,  $Q_{dv}$ , act primarily frontogenetic while the contribution from horizontal mixing,  $Q_{dh}$ , is negligible everywhere.

The conditions for two types of hydrodynamic instability are evaluated for the whole R100 domain: favourable conditions
for inertial (centrifugal) instability are found only along coastlines. There, anticyclonic eddies develop rapidly from along-coast currents if the coast is on the right hand side (looking downstream) and if the coastline is anticyclonically curved. During the first day of the R100 integration, symmetric instability is likely to occur in about 5 % of the model domain but within two days, the probability drops to <1 %. Parallel to the rapidly decreasing probability increases the geostrophic Richardson number *Rigeo*. While at the beginning of the integration, *Rigeo* < 1 in about 24 % of the R100 domain, this number decreases to about 4 % after one day, indicating that mixed-layer instability is the main process depleting the reservoir of potential energy.</p>

While anticyclonic eddies are generated solely along coastlines due to inertial instability, cyclonic eddies are found in the entire R100 domain, but preferably in those regions where the water depth is less than about 40 m. A special feature of the cyclones is their ability to absorbe internal waves and to sustain patches of continous upwelling for several days favouring plankton growth (Mahadevan, 2016). By contrast to mesoscale cyclones which pinch off from basin-scale fronts, submesoscale

cyclones are rolled-up streamers, similar to those observed by Klymak et al. (2016) on the north wall of the Gulf Stream. Hence, it may not be appropriate to denote these features "eddies" but "spirals" instead, as suggested by Munk et al. (2000).

A peculiar feature of the observed eddy shown in 18a are the cold spots along its periphery. It is conjectured that the spots are small upwelling cells probably driven by gravitational instability which links the submesoscale with the microscale and finally

5

leads to three-dimensional energy disspation. As it is extremely challenging to observe such features with in situ methods, attempts will be made to catch these spots with further downscaling to the O(1) m scale using non-hydrostatic models or Large Eddy Simulations. These numerical approaches would also serve to explore structures and processes in the interior of eddies in greater detail.

*Code and data availability.* The model code and the output of the ROMS runs presented in this article are available from the first author on request.

*Author contributions.* Reiner Onken implemented the double-nested ROMS model for the given region in the Baltic Sea and conducted the model runs. Ingrid M. Angel-Benavides and Burkard Baschek contributed the results from the SubEx experiment in the Southern California Counter Current, and Burkard Baschek coordinated the "Expedition Clockwork Ocean" and the corresponding modeling activities.

Competing interests. The authors declare that they have no conflicts of interest.

15 Acknowledgements. We thank Rüdiger Röttgers and Hajo Krasemann for the processing of Figs. 4b and ??, and Rainer Feistel who provided details of the Arkona and Bornholm Basins to us. Geoffrey Smith and George Marmorino, Naval Research Laboratory, provided infrared observations taken during the Submesoscale Experiments for comparison. The ICON-EU output was made available by the Thomas Bruns of the German Weather Service (DWD). The output of the HIROMB-BOOS model was downloaded from the Copernicus Marine Environment Monitoring Service, and the bathymetry data from the General Bathymetric Chart of the Oceans were provided by the British Oceanographic 20 Data Centre. The coastline data were obtained from NOAA (National Oceanic and Atmospheric Administration).

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

| Parameter         | Units                         | Value             | Meaning                                      |  |
|-------------------|-------------------------------|-------------------|----------------------------------------------|--|
| K                 | 1                             | 10                | number of vertical layers                    |  |
| $V_{tr}$          | N/A                           | 2                 | ransformation equation                       |  |
| $V_{str}$         | N/A                           | 1                 | stretching function                          |  |
| $	heta_s$         | 1                             | 5                 | surface control parameter                    |  |
| $	heta_b$         | 1                             | 0.4               | bottom control parameter                     |  |
| $h_c$             | m                             | 10                | critical depth                               |  |
| $A_V^T$           | $\mathrm{m}^2\mathrm{s}^{-1}$ | $4\times 10^{-5}$ | vertical mixing coefficient for tracers*     |  |
| $A_V^M$           | $\mathrm{m}^2\mathrm{s}^{-1}$ | $1\times 10^{-5}$ | vertical mixing coefficient for momentum*    |  |
| $\Delta t_{fast}$ | 1                             | 20                | Number of barotropic time-steps between each |  |
|                   |                               |                   | baroclinic time step                         |  |

Table 2. Parameter settings and properties of the R500 and R100 setups.

| Parameter/Property     | Units                         | R500                       | R100                       | Meaning                                           |
|------------------------|-------------------------------|----------------------------|----------------------------|---------------------------------------------------|
| $\Delta x$             | m                             | 500                        | 100                        | nominal zonal grid size                           |
| $\Delta y$             | m                             | 500                        | 100                        | nominal meridional grid size                      |
| $lon_{west}$           |                               | $13^{\circ}30'$ W          | $14^{\circ}12'~\mathrm{W}$ | western boundary of model domain                  |
| $lon_{east}$           |                               | $16^{\circ}30' \mathrm{W}$ | $15^{\circ}48'~\mathrm{W}$ | eastern boundary of model domain                  |
| $lat_{south}$          |                               | $53^{\circ}54'$ N          | $54^\circ 18'$ N           | southern boundary of model domain                 |
| $lat_{north}$          |                               | $55^{\circ}30'$ N          | $55^{\circ}12'$ N          | northern boundary of model domain                 |
| $N_x$                  | 1                             | 386                        | 1033                       | number of tracer grid points (zonal)              |
| $N_y$                  | 1                             | 356                        | 1004                       | number of tracer grid points (meridional)         |
| domain size            | $\mathrm{km}^2$               | $193\times178$             | $103\times100$             |                                                   |
| $\Delta t$             | s                             | 150                        | 30                         | baroclinic time step                              |
| $A_H^T$                | $\mathrm{m}^4\mathrm{s}^{-1}$ | $5\times 10^5$             | $10^{3}$                   | bi-harmonic eddy diffusivity coefficient          |
| $A_H^M$                | $\mathrm{m}^4\mathrm{s}^{-1}$ | $10^{3}$                   | N/A                        | bi-harmonic eddy viscosity coefficient            |
| $A_H^M$                | $\mathrm{m}^2\mathrm{s}^{-1}$ | N/A                        | $10^{-2}$                  | mono-harmonic eddy viscosity coefficient          |
| vertical mixing scheme | N/A                           | GLS                        | KPP                        |                                                   |
| $rx_0$                 | 1                             | 0.16                       | 0.07                       | stability condition after Haidvogel et al. (2000) |
| $rx_1$                 | 1                             | 1.75                       | 0.84                       | stability condition after Haney (1991)            |
|                        |                               |                            |                            |                                                   |