# Peer review of "Very high-resolution modelling of submesoscale turbulent patterns and processes in the Baltic Sea"

_Ocean Science, 2019_

## Referee Comment (RC1) · Anonymous Referee #1 · 14 Jun 2019

A review of "Very high-resolution modelling of submesoscale turbulent patterns and processes in the Baltic Sea" (authors Reiner Onken, Burkard Baschek, and Ingrid M. Angel-Benavides)

**Overall rating**

This is an interesting study aimed to simulate submesoscale patterns in the Baltic Sea and comprehensively discuss different aspects of the phenomenon. The paper can be eventually published after moderate revision.

There are several major remarks and a handful of minor ones and typos.

**Major remarks:**

1 In R100 the atmospheric forcing was turned off "to analyse the kinematic and dynamical properties of STPPs without disturbing effects". However, one would expect the that the STPPs generated without and with atmospheric forcing to be substantially different, while the goal of the study is to model STPPs in the Baltic Sea, that is, including all "disturbing effects" existing in reality. In view of the above, I'm not sure that e.g. the main features of the evolution of submesocsale eddy C3 shown in Fig. 10 will be reproduced by R100 with turned on atmospheric forcing?

2 The prognostic run of R500 started from initial and boundary conditions generated by not eddy-resolving HBM on June 1, 2016, and already in 15 days, on 15 June, the R100 was initialized from R500. The 15 day period does not seem long enough to provide a well-developed (populated with eddies) STPPs from not eddy-resolving initial fields. Very high-resolution modelling previously performed in the Baltic Sea (more specifically, in the Gulf of Finland) by Väli et al. (2017) showed that some cyclonic eddies that can be referred as submesoscale creatures in view of the relative vorticity well exceeding f, can live more for than a month. The only comparison of the simulated STPPs with satellite imagery for the modelled period showed that the observed cyclonic spiral, the most prominent feature of the Sentiel-3 image (Fig. 4, bottom) had rather sluggish counterpart in R500 and no counterpart in R100 (cf. Figs. 4 and 5). If the R500 started earlier, e.g. on May 1, the observed spiral would be probably reproduced more realistically/reliably. Since the submesoscale eddies can travel for a long distance (Väli et al., 2017) it seems preferable also to take the nested domain for R500 larger, e.g. including the whole Arkona and Bornholm basins.

3 The authors did not seem to be able to find any convincing link between the results of the field experiment "Expedition Clockwork Ocean" and the submesoscale modelling they carried out. The related pieces of text and drawing (Fig. 16)) could be dropped, which would make this long article easier to read.

4 It seems that the authors are not familiar with recent publications on STPPs modeling in the Baltic Sea (Väli et al., 2017, 2018). Meanwhile, based on a 0.125 nautical mile grid model of the Gulf of Finland, Baltic Sea, Väli et al. (2017; 2018) found submesoscale patterns of relative vorticity, absolute horizontal gradient of potential density and many other tracers similar to presented in this paper, so it would be nice to compare one with the other.

Citation:

Väli, G., V. Zhurbas, U. Lips, J. Laanemets, 2017. Submesoscale structures related to upwelling events in the Gulf of Finland, Baltic Sea (numerical experiments), J. Mar. Syst., 171(SI), 31–42.

Vali G., Zhurbas V.M., Laanemets J., Lips U., 2018. Clustering of floating particles due to submesoscale dynanics: a simulation study for the Gulf of Finland. Fundamentalnaya i prikladnaya gidrofizika, 11(2), 21-35, DOI: 10.7868/S2073667318020028 (open access at http://hydrophysics.info)

**Minor remarks**

P7L5 "*a high-salinity eddy in the Arkona Basin, and mushroom-like patterns east and southeast of Bornholm on 1 and 10 June, respectively*" There is no any high-salinity eddy in the Arkona Basin on 1 June when both HBM and R500/R500NF display the same not eddy-resolving pattern (see Fig. 2).

P7L26 "An analysis of the prognostic fields of R500\_NF yielded an unexpected finding: the tracer fields exhibit much more spatial variability in comparison to the corresponding fields of R500 (see the right panel in Fig. 2)" To my mind, it is a very expected finding: results of remote sensing (Kubryakov and Stanichny, 2015), modelling (Zhurbas et al., 2008; Väli et al., 2017), and even laboratory experiments (Zatsepin et al., 2005) showed that mesoscale/submesoscale structures begin to grow rapidly when the wind subsides.

**Citation:**

Kubryakov A.A., Stanichny S.V., 2015. Seasonal and interannual variability of the Black Sea eddies and its dependence on characteristics of the large-scale circulation, Deep-Sea Research I, 97, 80–91.

Zatsepin AG, Denisov ES, Emelyanov SV et al., 2005. Effect of bottom slope and wind on the near-shore current in a rotating stratified fluid: laboratory modeling for the Black Sea, Oceanology 45(Suppl 1): S13–S26.

Zhurbas, V., J. Laanemets, and E. Vahtera, 2008. Modeling of the mesoscale structure of coupled upwelling/downwelling events and the related input of nutrients to the upper mixed layer in the Gulf of Finland, Baltic Sea, J. Geophys. Res. - Oceans, 113, C05004.

P8L20. It seems worth to compare the tracer patterns of  $|\nabla \rho|$  and  $\zeta$  with that of Väli et al. (2018) simulated in the Gulf of Finland at 0.125 nautical mile grid.

P9L23 It seems worth to compare the relative vorticity statistics with that of Väli et al. (2017).

P10L28. "*The topography of potential density surfaces in the anticyclone shows that the patches are accompanied by large excursions of isopycnals, indicating intense internal wave activity.*" ROMS is a hydrostatic model which does not describe internal waves except for near-inertial waves that propagate almost vertically and therefore are hardly able to produce large vertical excursions of isopycnals at short horizontal scales of O(1km). Please comment the issue.

P17L19-22.  $Ro \sim O(1)$  and  $Ri \sim O(1)$  are mentioned as the criteria of submesoscale fronts, but in Fig. 14 the plot of *Ri* is missing (in contrast to the *Ro* plot).

P17L31. Fig. 15 is really a spectacular satellite image of a phytoplankton bloom but in the context of this article, it seems far-fetched because it was received at another time, in another place with other bottom topography, shoreline, stratification, currents, atmospheric forcing... The fact that the Rossby radius in this place is of the same order than that of the Bornholm and Arkona basins does not seem to be a serious legitimation. The authors did not model circulation off the Estonian coast and therefore have no information on whether *Ro* is large enough to attribute the spirals in Fig. 15 to submesoscale structures. I would suggest to drop Fig. 15 and the related piece of text.

P18L30. "Moreover, salinity was chosen for comparison because it is the primary component controlling the stratification in the Baltic Sea." There is some confusion here... That is true that in the whole the Baltic Sea stratification is controlled by salinity due to the presence of a lower layer filled with high salinity water of the North Sea origin. But in the upper layer of 60-m depth (i.e. above the permanent halocline), density stratification is primarily controlled by temperature, especially in Summer when the seasonal thermocline is developed. The 15-m depth salinity in Fig. 17 (right) displays ~0.1 psu excess in the C3 centre which contributes to density stratification as much as the temperature deficit of ~0.3°C, but one would expect that the actual temperature deficit is much larger, e.g. >1°C, and therefore the salinity is a secondary

component controlling the stratification in C3 (i.e. the salinity in C3 behaves like a passive tracer). To clarify the issue, please add the 15-m depth temperature to Fig. 17.

Table 2. Were  $A_H^T$ ,  $A_H^M$  [m4s-1], and  $A_H^M$  [m2s-1] really taken constants? Why the Smagorinsky parameterization was not applied?

**Technical corrections/typos**

P6L4. cyle->cycle

P11L3. Class number is missing.

P12L10. Two "are" in a row

P13L21. Two "is" in a row

P18L23. "spiraliform". Google Translator doesn't know such a word.

Table 1. The number of vertical layers is 10. This is a typo, isn't it?

Figs. 9, 10, 11, and 13. Scale for velocity vectors is missing.

---

## Referee Comment (RC2) · Anonymous Referee #2 · 20 Jun 2019

A nested numerical model is applied to a part of the Baltic Sea to analyse surface submesoscale dynamics. There are three nesting levels, using different models and model configurations, to forced the inner nest with a high spatial resolution of 100m.

For several reasons, this paper cannot be recommended for publication in Ocean Science:

1. The model nesting is inconsistent, with a daily forcing from the outer model and a bathymetry in the inner nests which is based on a coarse bathymetry reconstruction although high-resolution bathymetry data should be available for this region. Boundary data transfer from the out model to the inner model seems incomplete. The middle nest uses a high-order turbulence closure whereas the high-resolution nest uses a bulk method (the KPP model, which is not taking into account several relevant processes),

because the high-resolution method is computationally expensive.

2. The simulations in the high-resolution nest are carried out without wind forcing, because with wind forcing, the surface variability in salinity is much weaker than the observed ocean color variability. All analysis is based on this simulation without wind forcing.

3. There is no model validation carried out at all, although high-quality observational data should be available, as described in lines 8-14. The only qualitative comparison between model results and observations shows a snapshot of the simulated sea surface salinity (without wind forcing) and an ocean color image (representing chlorophyll). There is no evidence that those patterns should be similar.

Therefore, I recommend to reject this submission.

Some detailed comments to pages 1 – 8:

Page 1:

8: "high vertical velocities" instead of "strong vertical speeds"

11: What is the tendency equation?

18: delete "ambitious". All research should be ambitious.

19: The word "turbulent" is missing here.

20: Here, turbulent kinetic energy seems to be used with a different meaning than usual. TKE is generally understood as non-hydrostatic which cannot be reproduced by a hydrostatic model. Do you here mean eddy kinetic energy (EKE)?

22-24: Does this whole range of scales also occur at one location? I thought that sub-mesoscale turbulence is defined by non-dimensional parameters such as the Rossby number or the Richardson number (both being order of unity or larger). In the Bornholm Sea, you find probably not find submesoscale features of 100 m scale in the vertical.

Page 2:

1-5: I feel that this discussion of the many order of magnitude is an unnecessary overstatement. There is no turbulence on the 1000 km scale and I would base the definition on these non-dimensional numbers only. I propose that the Rossby number is defined here. For a Rossby number of order of unity, I would not say that Earth rotation is on minor importance. That should be the case only for much larger Rossby numbers.

6-14: The red cascade is in contradiction to the spontaneous emission of subme-soscale structured from the meso-scale. So, we indeed have a (blue) forward cascade. The contradiction is due to the fact that there are about 40 years between the results referenced here. Therefore, I recommend to reformulate this paragraph.

Page 4:

16: Explain what an HBM operational model is.

Page 5:

10: It is a bit surprising that daily mean fields are used to forced a model simulations which is designed to reproduce highly variable dynamics. The reasoning for this needs to be motivated.

18: It is unclear why different turbulence closure models have been used for the 500 m and 100 m models. And specifically confusing is that the large-scale closure model KPP is used for the very high-resolution simulation of 100m. There will be many relevant processes at that scale which are not reproduced by the KPP such as static instabilities induced by differential advection. I would also suspect that the KPP model does not reproduce the logarithmic bottom boundary layer.

27: The use of the s-coordinate should be mentioned in the paragraph above, along with the information about the number of vertical layers. I find it very unusual to base a high-performance model simulation on such a weak data basis. The boundary data

must include depth information, and with that the depth should be known.

Page 6:

30-32: This information about the turbulence closure model has partially been given before (page 5). The additional information given here should be moved up to section 2.3.

Page 7:

6-7: Length scales are partially given in metric units and partially in nautical miles. I recommend to use metric units.

8-9: Which gap is there between the meso-scale and the submeso-scale?

12: TKE for Total Kinetic Energy is very confusing, also since turbulent kinetic energy is used as well. I propose using KE for Total Kinetic Energy.

Figure 3: It might not be clear to everyone what "cumulatively averaged" TKE is. It should also be stated that this is the kinetic energy per unit mass.

17-25: It does not become clear, why the unforced simulation is carried out and why it has been performed. Either do not discuss it or make sure that it gives a clear message. I specifically do not see how the spin-up time of the 500m nest can be estimated by this procedure. Lines 30-32 may contain an insight from this exercise. But it is not clear why vertical mixing should enhance horizontal mixing that would blur the fine structures. And if that happens, it should probably be a relevant process that needs to be discussed here in depth.

P 7, 33 – p 8, 8: It is not clear why one quantity in the model simulations (salinity) is validated with another quantity (ocean color). The latter is certainly related to chlorophyll which in turn might be related to phytoplankton concentration, a possibly positively buoyant particulate matter. It is not clear, why these two quantities should be related. Why do you not compare observed surface temperature with simulated surface temperature?

10-19: I have read this section many times, and I am quite sure that you say that you carry out the 100m simulation without any surface forcing because it would blur the submesoscale structures. It that really true? It is well acknowledged in the literature that submesosales are driven by surface forcing as well. So, if the model results detoriate due to the surface forcing, then the consequence should not be shut off the forcing but to find the reason between model results and simulations. Something seems to be fundamentally wrong.

If I see it right, a numerical is analysed here of which the only single validation is the qualitative comparison between a snapshot of surface salinity and an ocean color snapshot. There is wind forcing in reality and no wind forcing in the model. Some of the structures look similar. That's all. Afterwards, the model results are intensively analysed in terms of submesoscale dynamics.

Clearly, that is not science.
* * *

---

## Author Comment (AC1) · 24 Jun 2019

Dear Reviewer#1,

Thank you very much for your comprehensive review of our manuscript. Please find below our replies to your comments. Note that below your comments are written in blue while our replies are black.

**A review of** "Very high-resolution modelling of submesoscale turbulent patterns and processes in the Baltic Sea" (authors Reiner Onken, Burkard Baschek, and Ingrid M. Angel-Benavides)
**Overall rating**
This is an interesting study aimed to simulate submesoscale patterns in the Baltic Sea and comprehensively discuss different aspects of the phenomenon. The paper can be eventually published after moderate revision.
There are several major remarks and a handful of minor ones and typos.

**Major remarks:**
1 In R100 the atmospheric forcing was turned off "to analyse the kinematic and dynamical properties of STPPs without disturbing effects". However, one would expect the that the STPPs generated without and with atmospheric forcing to be substantially different,

They are indeed substantially different, see below

while the goal of the study is to model STPPs in the Baltic Sea, that is, including all "disturbing effects" existing in reality.

You are right, the goal of this study is *to understand and and interprete the observed features by means of of high-resolution modeling* (see manuscript P1 L21). However, in order to *understand* the features, all "disturbing effects" have to be ignored for the time being. This is only possible in numerical models, where – in the present case – the atmospheric forcing was turned off in R100. Of course, this is a simplification, but simplifications are the strength of any model (not just numerical models) to understand complicated processes. Moreover, the impact of atmospheric forcing on the evolution of STPPs is in the works in a follow-up study.

Proposed action
We will add a corresponding remark in the Introduction and at the beginning of Section 4.

In view of the above, I'm not sure that e.g. the main features of the evolution of submesocsale eddy C3 shown in Fig. 10 will be reproduced by R100 with turned on atmospheric forcing. Could the authors present analogue of Fig. 10 with turned on atmospheric forcing?

Fortunately, the results of an R100 run with full atmospheric forcing are still available. We will first demonstrate the impact of atmospheric forcing by the corresponding equivalent of Fig. 6, labelled as Fig. 6EQUIV. A comparison reveals that the atmospheric forcing has a dramatic impact on all near-surface variables. Primarily, the patterns of relative vorticity and divergence are blurred and do not allow any meaningful interpretation.
We plotted as well the equivalent of Fig. 10 with atmospheric forcing, but the result was as expected: there is no submesoscale eddy C3 at all, at least not in the corresponding location. We will save ourselves the plots.

Proposed action
In order to demonstrate the impact of atmospheric forcing, it is suggested to add Fig. 6EQUIV in the revised manuscript.

2 The prognostic run of R500 started from initial and boundary conditions generated by not eddy-resolving HBM on June 1, 2016, and already in 15 days, on 15 June, the R100 was initialized from R500. The 15 day period does not seem long enough to provide a well-developed (populated with eddies) STPPs from not eddy-resolving initial fields.

We do not agree. According to Fig. 3 and P7 L22–25, the spin-up time of R500_NF was estimated to 12 days. And the right column of Fig. 2 confirms that already on 10 June the domain is populated with mesoscale meanders and eddies. The mesoscale activity then increases until 20 June but it remains more or less constant thereafter. Hence, 15 June is well suited to initialize R100. Please note that  R100 was initialized from R500 (not from R500_NF) which did not provide a realistic estimate for the spin-up period.

Very high-resolution modelling previously performed in the Baltic Sea (more specifically, in the Gulf of Finland) by Väli et al. (2017) showed that some cyclonic eddies that can be referred as submesoscale creatures in view of the relative vorticity well exceeding f, can live more for than a month. The only comparison of the simulated STPPs with satellite imagery for the modelled period showed that the observed cyclonic spiral, the most prominent feature of the Sentiel-3 image (Fig. 4, bottom) had rather sluggish counterpart in R500

Why do you say that the spiral is "sluggish" in R500? In Fig. 4, we compared tracer patterns at the surface which do not provide any information about the magnitude of currents.

and no counterpart in R100 (cf. Figs. 4 and 5).

Unfortunately, on 23 June in Fig. 5 are shown only salinity (left column) and temperature (middle) while velocity vectors were omitted. In Fig. 5EQUIV are plotted the same variables for a zoomed area but with vectors of the near-surface velocity superimposed. Those vectors indicate clearly the centre of the cyclone at about the same position as in Fig. 4a. Hence, there *is* a counterpart in R100.

Proposed action
Horizontal velocity vectors might be superimposed to salinity and temperature on 23 June in Fig. 5, however, that might make the figure unreadable. Instead, this issue will be discussed in the text.

If the R500 started earlier, e.g. on May 1, the observed spiral would be probably reproduced more realistically/reliably. Since the submesoscale eddies can travel for a long distance (Väli et al., 2017) it seems preferable also to take the nested domain for R500 larger, e.g. including the whole Arkona and Bornholm basins.

In our opinion, the issue with the spiral is solved (see above) – isn't it? Please note that other observed features of Fig. 4b (C2 and the fronts in the NW corner) are reasonably well reproduced in Fig. 4a. We doubt that the suggested earlier start of R500 would lead to a significant improve of the model results.
On the other hand, a start of R500 on 1 May would mean to redo the entire manuscript, including all the graphics. Moreover, no atmospheric forcing is available for May; we would have to purchase it from DWD.

Proposed action
none

3 The authors did not seem to be able to find any convincing link between the results of the field experiment "Expedition Clockwork Ocean" and the submesoscale modelling they carried out. The related pieces of text and drawing (Fig. 16) could be dropped, which would make this long

article easier to read.

Most of the work for this article was done in 2017/2018. A that time, tangible observational results from "Expedition Clockwork Ocean" were not yet available.

Proposed action
We will drop Fig. 16 and the related pieces of text.

4 It seems that the authors are not familiar with recent publications on STPPs modeling in the Baltic Sea (Väli et al., 2017, 2018). Meanwhile, based on a 0.125 nautical mile grid model of the Gulf of Finland, Baltic Sea, Väli et al. (2017; 2018) found submesoscale patterns of relative vorticity, absolute horizontal gradient of potential density and many other tracers similar to presented in this paper, so it would be nice to compare one with the other.
Citation:
Väli, G., V. Zhurbas, U. Lips, J. Laanemets, 2017. Submesoscale structures related to upwelling events in the Gulf of Finland, Baltic Sea (numerical experiments), J. Mar. Syst., 171(SI), 31–42.
Vali G., Zhurbas V.M., Laanemets J., Lips U., 2018. Clustering of floating particles due to submesoscale dynanics: a simulation study for the Gulf of Finland. Fundamentalnaya i prikladnaya gidrofizika, 11(2), 21-35, DOI: 10.7868/S2073667318020028 (open access at http://hydrophysics.info)

Thank you very much for your tip! We have downloaded the above articles.

Proposed action
The desired comparison will be done.

**Minor remarks**
P7L5 "a high-salinity eddy in the Arkona Basin, and mushroom-like patterns east and southeast of Bornholm on 1 and 10 June, respectively" There is no any high-salinity eddy in the Arkona Basin on 1 June when both HBM and R500/R500NF display the same not eddy-resolving pattern (see Fig. 2).

You are right!

Proposed action
The corresponding sentences will be rephrased.

P7L26 "An analysis of the prognostic fields of R500_NF yielded an unexpected finding: the tracer fields exhibit much more spatial variability in comparison to the corresponding fields of R500 (see the right panel in Fig. 2)" To my mind, it is a very expected finding: results of remote sensing (Kubryakov and Stanichny, 2015), modelling (Zhurbas et al., 2008; Väli et al., 2017) , and even laboratory experiments (Zatsepin et al., 2005) showed that mesoscale/submesoscale structures begin to grow rapidly when the wind subsides.

You are right!

Proposed action
The corresponding sentences will be rephrased; missing references will be included.

Citation:
Kubryakov A.A., Stanichny S.V., 2015. Seasonal and interannual variability of the Black Sea eddies and its dependence on characteristics of the large-scale circulation, Deep-Sea Research I,

97, 80–91.

No open access; ordered via library

Zatsepin AG, Denisov ES, Emelyanov SV et al., 2005. Effect of bottom slope and wind on the near-shore current in a rotating stratified fluid: laboratory modeling for the Black Sea, Oceanology 45(Suppl 1): S13–S26.

Not available online. Full text requested via Research Gate

Zhurbas, V., J. Laanemets, and E. Vahtera, 2008. Modeling of the mesoscale structure of coupled upwelling/downwelling events and the related input of nutrients to the upper mixed layer in the Gulf of Finland, Baltic Sea, J. Geophys. Res. - Oceans, 113, C05004.

P8L20. It seems worth to compare the tracer patterns of $|\nabla \rho|$ and $\zeta$ with that of Väli et al. (2018) simulated in the Gulf of Finland at 0.125 nautical mile grid.

Proposed action
To be done

P9L23 It seems worth to compare the relative vorticity statistics with that of Väli et al. (2017).

Proposed action
To be done

P10L28. "The topography of potential density surfaces in the anticyclone shows that the patches are accompanied by large excursions of isopycnals, indicating intense internal wave activity." ROMS is a hydrostatic model which does not describe internal waves except for near-inertial waves that propagate almost vertically and therefore are hardly able to produce large vertical excursions of isopycnals at short horizontal scales of O(1km). Please comment the issue.

It is known to the authors that internal waves are insuffienctly reproduced in hydrostatic models. Therefore, we just started to investigate this issue in greater detail in our group with CROCO, a non-hydrostatic version of ROMS_AGRIF.

Proposed action
The corresponding passages in the manuscript will be rephrased.

P17L19-22. Ro~O(1) and Ri~O(1) are mentioned as the criteria of submesoscale fronts, but in Fig. 14 the plot of Ri is missing (in contrast to the Ro plot).

You are right!

Proposed action
We will add a plot of the Richardson number.

P17L31. Fig. 15 is really a spectacular satellite image of a phytoplankton bloom but in the context of this article, it seems far-fetched because it was received at another time, in another place with other bottom topography, shoreline, stratification, currents, atmospheric forcing... The fact that the Rossby radius in this place is of the same order than that of the Bornholm and Arkona basins does not seem to be a serious legitimation. The authors did not model circulation off the Estonian coast and therefore have no information on whether Ro is large enough to

attribute the spirals in Fig. 15 to submesoscale structures. I would suggest to drop Fig. 15 and the related piece of text.

Proposed action
Fig. 15 will be dropped

P18L30. "Moreover, salinity was chosen for comparison because it is the primary component controlling the stratification in the Baltic Sea." There is some confusion here... That is true that in the whole the Baltic Sea stratification is controlled by salinity due to the presence of a lower layer filled with high salinity water of the North Sea origin. But in the upper layer of 60-m depth (i.e. above the permanent halocline), density stratification is primarily controlled by temperature, especially in Summer when the seasonal thermocline is developed. The 15-m depth salinity in Fig. 17 (right) displays ~0.1 psu excess in the C3 centre which contributes to density stratification as much as the temperature deficit of ~0.3°C, but one would expect that the actual temperature deficit is much larger, e.g. >1°C, and therefore the salinity is a secondary component controlling the stratification in C3 (i.e. the salinity in C3 behaves like a passive tracer). To clarify the issue, please add the 15-m depth temperature to Fig. 17.

Proposed action
The 15-m temperature was added (see Fig. 17EQUIV). The figure caption and the corresponding text passages will be modified accordingly.

Table 2. Were $A^T_H$ , $A^M_H$ [m$^4$s$^{-1}$], and $A^M_H$ [m$^2$s$^{-1}$] really taken constants? Why the Smagorinsky parameterization was not applied?

We do not really understand your question. $A^T_H$ , $A^M_H$, and $A^M_H$ are the *coefficients* of the bi-harmonic and mono-harmonic formulation for horizontal diffusion and horizontal eddy viscosity, where the bi-harmonic formulation is – to our knowledge – the so-called Smagorinsky parameterization.
Both formulations were tested with various coefficients, and those listed in Table 2 were considered to be the best job in the sense "minimum diffusion but maximum damping of numerical (2-deltax) noise".
Are we wrong?

**Technical corrections/typos**
P6L4. cyle→cycle                          OK
P11L3. Class number is missing.           OK
P12L10. Two "are" in a row                OK
P13L21. Two "is" in a row                 OK
P18L23. "spiraliform" . Google Translator doesn't know such a word.
            But dict.leo.org (German ← → Englisch) does!
Table 1. The number of vertical layers is 10. This is a typo, isn't it?
            No – there are K=10 layers; what seems to be the trouble?
Figs. 9, 10, 11, and 13. Scale for velocity vectors is missing.
            The plots were done with the MATLAB function *m_quiver.m*
            We will try to add a scale.

We hope that our proposed actions satisfy your criticism!

Best regards,
Reiner Onken and co-authors

[Figure]

*Figure 6EQUIV: Equivalent of Fig. 6 but with full atmospheric forcing.*

[Figure]

*Fig. 5EQUIV: Top-layer salinity, temperature and horizontal velocity on 23 June in a subarea of R100. The magenta arrow points to the centre of the cyclonic spiral.*

[Figure]

*Fig. 17EQUIV:*

---

## Referee Comment (RC3) · Anonymous Referee #1 · 27 Jun 2019

Dear Authors, For unknown reasons, I failed to save your os-2019-44-AC1-supplement.pdf in Word-docx format for further insertion of my comments. Therefore, I have nothing left but to insert the comments into the pdf in the form of sticky notes (see attached os-2019-44-AC1-supplement_Reply.pdf). Kind regards, Reviewer #1

Please also note the supplement to this comment:
https://www.ocean-sci-discuss.net/os-2019-44/os-2019-44-RC3-supplement.pdf
* * *

---

## Referee Comment (RC4) · Anonymous Referee #3 · 18 Jul 2019

Onken et al. Very high-resolution modelling of submesoscale turbulent patterns and processes in the Baltic Sea; Ocean Sci. Discuss., https://doi.org/10.5194/os-2019-44

Review

Major comments

The authors present an interesting numerical experiment for studying the evolution and characteristics of submesoscale processes. It is worth to be published when appropriately revised, especially in regard to the interpretation of results.

My major recommendation is that the manuscript has to be rewritten to state clearly the aim of the study and the methods applied and to interpret the results accordingly. Authors present a numerical experiment where atmospheric forcing was turned off for

the model with the highest resolution while the initial and boundary conditions were created with the model with atmospheric forcing. It is unclear what we could learn from such an experiment regarding submesoscale processes. Is the aim to show how the submesoscale processes evolve in case of the sudden vanishing of external forcing and what are their characteristics in such conditions?

The other concern is related to the description of methods. You state that special care was taken for the preparation of initial and boundary conditions for the R100 model as the nesting ratio of five is challenging. However, you do not provide any arguments on why the cubic spline was used. Did you try other methods or run any sensitivity tests? Are some results (as the revealed false patterns) somehow related to this procedure? No information is given about the atmospheric forcing in HBM. It is not clear how the domain averaged TKE and cumulative averaged TKE are calculated, etc. See the specific comments below.

Specific comments

P1L8: Are the speeds of 100 m day-1 characteristic for submesoscale processes or internal waves?

P1L10-12: Sentences like "The conditions for inertial and symmetric instability are evaluated for the whole domain, and the components of the tendency equation are computed in a subregion." are not informative in the concluding section of the abstract.

P5L9-10: Why HBM daily mean fields were used? What about the atmospheric forcing for HBM?

P6L4-5: What is meant by the term "cycle"? How can the whole first cycle be used as initial conditions?

P6L12-13: The use of cubic spline is not justified. Did you run any tests how the interpolation method could influence the results?

P6L21-23: Atmospheric forcing for R500 and R100 is mentioned here for the first time.

What about atmospheric forcing in HBM? Do the forcing sources differ between HBM and ROMS? Mentioning of atmospheric forcing and interpolation of forcing parameters for R100 is not relevant since you present the results from runs without atmospheric forcing.

P7L1-2: I do not understand this statement about salinity as the ideal parameter. Your atmospheric forcing has a resolution of 6.5 km. What could cause the blurring of the surface signal of temperature in submesoscale range when using such forcing?

P7L5-6: No difference between the HBM and R500 on 1 June – it is the initial day when the fields are identical as seen in Fig. 2.

P7L12-13: How the domain averaged TKE is calculated (all model layers, volume average)?

P7L13-14: What is "the cumulative average TKER500"?

P7L15-25: The aim of the entire section is unclear. You wrote that the idea was to provide a rough estimate of the spin-up time, but you discuss something else. It is a numerical experiment to show what happens if you use the initial and boundary conditions with atmospheric forcing and finer model domain where the forcing is turned off.

P7L26-32: From this, it is clear that the results presented later as R100 outcome are non-realistic (these are the results of an artificial numerical experiment; no point to compare the results directly with measurements).

P7L33-P8L8: This qualitative/visual comparison is OK, but I would not recommend to focus on it too much. A question would be whether R500 did reproduce the pattern qualitatively better than HBM, for instance.

P8L10-15: It is a numerical experiment, where the forcing is turned off, and the aim could not be to reproduce the observed fields. A comparison with the measurements would be feasible only if forcing is on.

P8L22: A very thin surface layer is picked for the presented maps. Could it be possible to show the same maps, for instance, at 5 m depth?

P9L2-5: Is this effect caused by the fact that you have initial and boundary conditions taken from a model output with forcing and the maps presented are from a sub-region without forcing? It is not clear a priori what is related to the natural variability and what to the model set-up.

P9L13-14: In addition to dates, also the time should be referred (these figures are not daily average fields, I suppose).

P9L16-22: The same question as above – you should interpret the data as outcomes of a numerical experiment.

P10L9-13: What that means? Could cubic spline cause such structures in the derivatives of the fields?

P10L23-25 and P11L6-7: Could you reveal the period of these internal waves?

P11L3: Which Class?

P12L12: Fig. 9 top row is referred to as the vertical velocity at 5m depth, but in Fig. 9c the vertical velocity at the base of the top layer is presented.

P12L29-30: Could the used cubic spline create more structures with the length scale of 1 km (R500 has the resolution of 500m)? Such a scale is well visible in many figures.

P12L31: Correct to "bimodal". What are the three-modal structures?

P13L25: Could such false advection effects influence the results (statistics) in general as well?

P15L15-16 and P15L25-26: How these findings of fast changes during first days of integration can be interpreted? Are these caused by the fact that the initial fields were taken from a model output with atmospheric forcing and R100 was run without atmospheric forcing?

P17L26-28: Please, give references.

P17L29-30: It is true, if you mean 2D distributions. However, there are publications based on ferrybox and glider measurements covering large areas and presenting statistics of submesoscale variability (even in the Baltic).

P17L32-P18L1: What do you mean by "hydrodynamical instability" here?

P18L9-10: Please, give references.

P18L14-15: Such 10m spots are not relevant in this context.

P18L21: Also Fig. 17 is not directly necessary to be presented.

P18L30-31: It is true in autumn-winter, but not in summer in the upper layer of the Baltic where the seasonal thermocline develops.

P20L8-10: This is a crucial point that the forcing was turned off in the high-resolution model domain, but it was still turned on in the model from where the initial and boundary conditions were extracted.

P21L3-4: Is this statement about cyclonic eddies favoring the plankton growth a result of the present study? You could insert a reference.

Tables

Table 1: I hope the number of vertical layers is more than 10.

Figures

Fig. 2: What is meant by cumulatively averaged TKE? Where the wind data come from?

Fig. 5: Could you add time (is it 0:00)?

Fig. 6: Time reference is missing.

Fig. 8: Density anomaly values could be given.

Fig. 11: Check the caption for English.

Fig. 15: What is the location of this image? Is it relevant here?

Fig. 16: It is something else than discussed in the manuscript. Consider dropping this figure.

Fig. 17. I am not sure how relevant this figure is, especially since it is from the ocean while the paper is about the Baltic where the scales are different.

[Figure]

---

## Author Comment (AC2) · 2 Aug 2019

Dear Reviewer#1,

Thank you very much for your comprehensive review of our manuscript. Please find below our replies to your comments. Note that below your comments are written in blue while our replies are black.
* * *
**Reply to your comments of 14 June (os-2019-44-RC1-supplement.pdf)**

\_\_\_\_\_

(1) A review of "Very high-resolution modelling of submesoscale turbulent patterns and processes in the Baltic Sea" (authors Reiner Onken, Burkard Baschek, and Ingrid M. Angel-Benavides) **Overall rating**

This is an interesting study aimed to simulate submesoscale patterns in the Baltic Sea and comprehensively discuss different aspects of the phenomenon. The paper can be eventually published after moderate revision.

There are several major remarks and a handful of minor ones and typos.

**Major remarks:**

1 In R100 the atmospheric forcing was turned off "to analyse the kinematic and dynamical properties of STPPs without disturbing effects". However, one would expect the that the STPPs generated without and with atmospheric forcing to be substantially different,

They are indeed substantially different.

Action see below (2), (3), (11)

**(2) while the goal of the study is to model STPPs in the Baltic Sea, that is, including all "disturbing effects" existing in reality.**

You are right, the goal of this study is *to understand and and interprete the observed features by means of of high-resolution modeling* (see manuscript P1 L21). However, in order to *understand* the features, all "disturbing effects" have to be ignored for the time being. This is only possible in numerical models, where – in the present case – the atmospheric forcing was turned off in R100. Of course, this is a simplification, but simplifications are the strength of any model (not just numerical models) to understand complicated processes. Moreover, the impact of atmospheric forcing on the evolution of STPPs is in the works in a follow-up study.

**Action**

- New piece of text in the Abstract (new ms P1L7)
- Rewritten Introduction (new ms P4L8-12)
- New piece of text in Section 4 (new ms P8L26-28)
- New Section 4.3 "Impact of atmospheric forcing" (new ms P12L14-33)

(3) In view of the above, I'm not sure that e.g. the main features of the evolution of submesocsale eddy C3 shown in Fig. 10 will be reproduced by R100 with turned on atmospheric forcing. Could the authors present analogue of Fig. 10 with turned on atmospheric forcing?

The impact of atmospheric forcing is demonstrated by the new Fig. 8 which is the equivalent of Fig. 7, but with full atmospheric forcing. A comparison reveals that the atmospheric forcing has a dramatic impact on all near-surface variables.

We plotted as well the equivalent of Fig. 10 with atmospheric forcing, but the result was as expected: there was no more a submesoscale eddy C3 at all, at least not in the corresponding location. We will save ourselves the plots.

**Action**

- New Fig. 8
- Related text added in new Section 4.3 (new ms P12L14-33)

(4) 2 The prognostic run of R500 started from initial and boundary conditions generated by not eddy-resolving HBM on June 1, 2016, and already in 15 days, on 15 June, the R100 was initialized from R500. The 15 day period does not seem long enough to provide a well-developed (populated with eddies) STPPs from not eddy-resolving initial fields.

We do not agree. According to Fig. 3 and P7L22–25, the spin-up time of R500\_NF was estimated to 12 days. And the right column of Fig. 2 confirms that already on 10 June the domain is populated with mesoscale meanders and eddies. The mesoscale activity then increases until 20 June but it remains more or less constant thereafter. Hence, 15 June is well suited to initialize R100. Please note that R100 was initialized from R500 (not from R500\_NF) which did not provide a realistic estimate for the spin-up period.

**Action: none**

(5) Very high-resolution modelling previously performed in the Baltic Sea (more specifically, in the Gulf of Finland) by Väli et al. (2017) showed that some cyclonic eddies that can be referred as submesoscale creatures in view of the relative vorticity well exceeding f, can live more for than a month. The only comparison of the simulated STPPs with satellite imagery for the modelled period showed that the observed cyclonic spiral, the most prominent feature of the Sentiel-3 image (Fig. 4, bottom) had rather sluggish counterpart in R500

Why do you say that the spiral is "sluggish" in R500? In Fig. 4, we compared tracer patterns at the surface which do not provide any information about the magnitude of currents. Moreover, as can be seen in Fig. 5UV below, the magnitude of the currents at the western flank of that spiral exceeds 15 cm/s which is about 3 times larger than the background current

Fig. 5UV: Top-layer horizontal velocity on 23 June in a subarea of R100.

**Action: none**

**(6) and no counterpart in R100 (cf. Figs. 4 and 5).**

Unfortunately, on 23 June in Fig. 5 are shown only salinity (left column) and temperature (middle) while velocity vectors were omitted. In Fig. 5EQUIV (below) are plotted the same variables for a zoomed area but with vectors of the near-surface velocity superimposed. Those vectors indicate clearly the centre of the cyclone at about the same position as in Fig. 4a. Hence, there *is* a counterpart in R100.

---

## Author Comment (AC3) · 2 Aug 2019

Dear Reviewer#3,

Thank you very much for your comprehensive review of our manuscript. Please find below our replies to your comments. Note that your comments are written in blue while our replies are black.

**Reply to your comments of 18 July (os-2019-44-RC4.pdf)**
* * *
**Major comments**

The authors present an interesting numerical experiment for studying the evolution and characteristics of submesoscale processes. It is worth to be published when appropriately revised, especially in regard to the interpretation of results.

(1)My major recommendation is that the manuscript has to be rewritten to state clearly the aim of the study and the methods applied and to interpret the results accordingly. Authors present a numerical experiment where atmospheric forcing was turned off for the model with the highest resolution while the initial and boundary conditions were created with the model with atmospheric forcing. It is unclear what we could learn from such an experiment regarding submesoscale processes. Is the aim to show how the submesoscale processes evolve in case of the sudden vanishing of external forcing and what are their characteristics in such conditions?

**See below (8c), (13), (14), (16), (18), (20), (28), (36)**

Please note as well the new Section 4.3 "Impact of atmospheric forcing" and the new Fig. 8.

(2) The other concern is related to the description of methods. You state that special care was taken for the preparation of initial and boundary conditions for the R100 model as the nesting ratio of five is challenging. However, you do not provide any arguments on why the cubic spline was used. Did you try other methods or run any sensitivity tests? Are some results (as the revealed false patterns) somehow related to this procedure? No information is given about the atmospheric forcing in HBM. It is not clear how the domain averaged TKE and cumulative averaged TKE are calculated, etc. See the specific comments below.

**Action: See below (5b), (7), (8a,b), (11), (12), (21), (25), (39)**

**Specific comments**

(3)P1L8: Are the speeds of 100 m day-1 characteristic for submesoscale processes or internal waves?

The horizontal scales of submesoscale processes span the range between 10 m and 10 km and time scales from hours to days. Therefore, internal gravity waves (IGWs) themselves are a submesoscale process besides fronts, eddies, filaments, and frontogenesis. Each of these processes creates strong vertical motions and it makes no sense to sort out the contribution of either process to the amplitude spectrum of the vertical velocity.

**Action: none**

(4)P1L10-12: Sentences like "The conditions for inertial and symmetric instability are evaluated for the whole domain, and the components of the tendency equation are computed in a subregion." are not informative in the concluding section of the abstract.

An abstract summarizes the major aspects of the entire paper, including (i) the overall purpose of the study and the research problem, (ii) the design and the methods, and (iii) major findings. The above mentioned sentence belongs clearly to item (ii) and should not be omitted.

**Action: none**

**(5a)P5L9-10: Why HBM daily mean fields were used?**

The task of R500 was to provide a realistic mesoscale environment for the 100-m nest. As the shortest time scales of mesoscale processes are on the order of days, the daily averaged fields of HBM were considered to be sufficient for driving R500 at the open boundaries. Moreover, the data volume of the daily averaged fields is around 600 MB for the entire month of June 2016, while the hourly fields would comprise close to 15 GB. That would clearly overschoot the mark.

**Action: none**

**(5b)P5L9-10: What about the atmospheric forcing for HBM?**

HBM is run operationally at DMI and BSH, where the BSH run is the backup run in case the DMI run fails. While the DMI run used DMI-HIRLAM forcing at 5 km resolution, COSMO-EU forcing at 7 km is applied with the BSH run. We could easily provide this information in the manuscript, but in our opinion it is not relevant because we used the HBM output *as is*.

**Action: none**

**(6)**P6L4-5: What is meant by the term "cycle"? How can the whole first cycle be used as initial conditions?**

The HBM daily mean output at CMEMS is provided in NetCDF files which contain all prognostic variables, i.e. the 3D fields T, S, U, V, and the 2D field SSH. For the initialization of R500 and the provision of the open boundary conditions, we downloaded those fields for the entire month of June 2016 in one single file. Hence, this file contained the prognostic variables at 30 time levels on 1, 2, 3, ..., 30 June. The data set for each time level is what we call a "cycle".

The first cycle contains the prognostic fields of 1 June, and that is used for the initialization of R500. From the remaining cycles of 2, 3, 4, ..., 30 June we used only the prognostic fields along the open boundaries of the R500 domain.

**Action:**

In order to avoid any misunderstanding, we renamed "cycle" to "record". See

• new manuscript P5L5,31,32

**(7)P6L12-13: The use of cubic spline is not justified. Did you run any tests how the interpolation method could influence the results?**

In preliminary test runs with linear interpolation, the relative vorticity of jet flows into the R100 domain (i.e. the flow across the open boundaries) looked unrealistic, because the width of cyclonic and anticyclonic shear zones was frequently the same. This is contrary to experience where the width of the cyclonic shear zone is narrower than that of the anticyclonic shear. Using spline interpolation, those jets looked more realistic even though not perfect. The latter applies predominantly to strong jets exceeding 20 cm/s, see new Fig. 7a,b: (i) the relative vorticity is unrealistic in the northwest corner and along the northern boundary of the domain. (ii) the tiny streak with anticyclonic vorticity < -1 at the eastern boundary at 55° 25' N is unrealistic. The above issue is an intrinsic problem of downscaling. We have discussed it with other modelers but a perfect solution is not yet available.

Action: A piece of text was added to Section 2.4. See new ms P6L5-12

**(8a)**P6L21-23: Atmospheric forcing for R500 and R100 is mentioned here for the first time.What about atmospheric forcing in HBM?

Action: see above (5b)

(8b)Do the forcing sources differ between HBM and ROMS?

Yes

Action: none

(8c)Mentioning of atmospheric forcing and interpolation of forcing parameters for R100 is not relevant since you present the results from runs without atmospheric forcing.

Action: As in the revised version, we refer also to an R100 run with atmospheric forcing, the corresponding piece of text in Section 2.4 was modified. See P6L19

**(9)**P7L1-2: I do not understand this statement about salinity as the ideal parameter. Your atmospheric forcing has a resolution of 6.5 km. What could cause the blurring of the surface signal of temperature in submesoscale range when using such forcing?

It is true that the resolution of the atmospheric forcing is 6.5 km. Hence, the downward shortwave radiation and the infrared back radiation from the clouds are almost (*almost* because of the cubic spline interpolation) identical everywhere in an area of 6.5 km  $\times$  6.5 km. However, this is not true for the other components of the heat budget, i.e. the longwave radiation flux from the ocean, sensible and latent heat fluxes. These quantities depend on the sea surface temperature SST, and the SST in turn is affected by the 3D velocity field and vertical mixing which are subject to submesoscale spatial variability. Therefore, the net surface heat flux varies on the same spatial scales and may blur the submesoscale SST distribution.

In contrast, the surface salinity in the 6.5 km  $\times$  6.5 km area is controlled by the net freshwater flux, i.e. by precipitation and evaporation. Here, precipitation is identical everwhere because it is not impacted by any ocean properties, while evaporation depends largely on atmospheric parameters like wind speed, relative humidity, and air pressure, but also on the air-sea temperature difference. The latter exhibits submesoscale spatial variability, but the impact on the surface salinity is negligible because the evaporated freshwater causes an increase of the surface salinity that is rapidly distributed within the mixed layer. It can be shown by a back-of-the-envelope calculation that under realistic conditions (evaporation rate ~ 1 mm/day, mixed-layer depth ~ 10 m), the corresponding salinity change is on the order of 10-3. This may blur the submesoscale surface salinity distribution but the effect is merely detectable.

Action: We have added an explanation in Section 3 (new ms P6L31-P7L5).

**(10)**P7L5-6: No difference between the HBM and R500 on 1 June – it is the initial day when the fields are identical as seen in Fig. 2.

Action: Sentence changed (new ms P7L6-8)

**(11)**P7L12-13: How the domain averaged TKE is calculated (all model layers, volume average)?

TKE is read directly from the diagnostics in the ROMS logfile. Units are [Energy]/[Mass]=Nm/kg=kg m s-2 kg-1=m s-2

Action: A definition was added (new ms P7L14-15)

(12)P7L13-14: What is "the cumulative average TKER500"?

In a *cumulative moving average* or just *cumulative average*, the data arrive in an ordered datum stream, and the user would like to get the average of all of the data up until the current datum point (see https://en.wikipedia.org/wiki/Moving\_average). Example: Assume we have a time series of n values x1, x2, x3, ..., xn. The first element C(1) of the cumulative average is C(1)=x1 The second element is C(2)=(x1+x2)/2The third element is C(3)=(x1+x2+x3)/3

The n-th element is

C(n)=(x1+x2+x3+...xn)/n

Action:

- A definition of the cumulative average was added (new ms P7L24-25)
- In order to avoid confusion, TKE was replaced by KE, because the acronym TKE is frequently used for *turbulent* kinetic energy.

**(13)**P7L15-25: The aim of the entire section is unclear. You wrote that the idea was to provide a rough estimate of the spin-up time, but you discuss something else. It is a numerical experiment to show what happens if you use the initial and boundary conditions with atmospheric forcing and finer model domain where the forcing is turned off.

Action: We have rewritten the entire paragraph (new ms P7L12-28)

**(14)**P7L26-32: From this, it is clear that the results presented later as R100 outcome are non-realistic (these are the results of an artificial numerical experiment; no point to compare the results directly with measurements).

Action: The entire paragraph has been rewritten (new ms P7L29-P8L8)

(15)P7L33-P8L8: This qualitative/visual comparison is OK, but I would not recommend to focus on it too much. A question would be whether R500 did reproduce the pattern qualitatively better than HBM, for instance.

**Good advice!**

Action: HBM and R500 are compared in the new ms P8L19-21

(16)P8L10-15: It is a numerical experiment, where the forcing is turned off, and the aim could not be to reproduce the observed fields. A comparison with the measurements would be feasible only if forcing is on.

Action: To make it clearer, a piece of text was added (new ms P8L26-28))

**(17)**P8L22: A very thin surface layer is picked for the presented maps. Could it be possible to show the same maps, for instance, at 5 m depth?

Below are shown salinity, potential temperature and the absolute horizontal density gradient on 26 June at 5-m depth. Except for that the width of frontal zones is wider, these images resemble closely the corresponding images in Fig. 5.

**Action: none**

(18)P9L2-5: Is this effect caused by the fact that you have initial and boundary conditions taken from a model output with forcing and the maps presented are from a sub-region without forcing? It is not clear a priori what is related to the natural variability and what to the model set-up.

We have repeated R100, but using R500\_NF for the initialization and the boundary conditions. The frequency distribution of  $|\nabla \rho|$  is shown in Fig. B below, while the distribution of the run described in the manuscript is shown in Fig. A. One can see:

- (1) On 15 June (=initial conditions), there are more high-gradient areas in B, or in other words, the gradients are stronger in B. This is plausible, because the atmospheric forcing in R500 has blurred the gradients.
- (2) Same on 20 June
- (3) Distributions on 25 June in A resembles closely the distribution on 20 June in B.
- (4) 29 June: more high-gradient areas in A (in contrast to 15 June)
- (5) In A, the frequency of strong gradients increases until 25 June. Thereafter, it decreases.

(6) In B, the frequency of strong gradients increases until 20 June. Thereafter, it decreases. Hence, according to (5) and (6), the "frontal arrest" occurs apparently when the strong gradients reach a critical value. Thereafter, the strong gradients become weaker. The critical value in A is reached on 25 June, while in B already on 20 June. It is not clear whether physical processes or numerical diffusion (or both) limit the increase of gradients. For the physical processes, Qw, the straining deformation by the vertical velocity (see equation (7)), would be a suitable candidate. However, that would require to compute time series of the components of the tendency equation which is beyond the scope of this paper.

Action:

- These new aspects are discussed in the new ms P9L20-P10L2 and in P21L8-9
- We have added new Fig. 6

---

## Author Comment (AC4) · 5 Aug 2019

A nested numerical model is applied to a part of the Baltic Sea to analyse surface submesoscale dynamics. There are three nesting levels, using different models and model configurations, to forced the inner nest with a high spatial resolution of 100m. For several reasons, this paper cannot be recommended for publication in Ocean Science:

1. The model nesting is inconsistent, with a daily forcing from the outer model and a bathymetry in the inner nests which is based on a coarse bathymetry reconstruction although high-resolution bathymetry data should be available for this region.

Both for R500 and R100, the GEBCO_2014 grid with 30 arc seconds resolution was used. This is the highest-resolution available bathymetry data set for the Baltic. See P5L29 of the original manuscript.

Boundary data transfer from the out model to the inner model seems incomplete. The middle nest uses a high-order turbulence closure whereas the high-resolution nest uses a bulk method (the KPP model, which is not taking into account several relevant processes), because the high-resolution method is computationally expensive.

We used indeed the KPP model for the R100 nest, because the usage of GLS would have doubled the CPU time from 2.5 to about 5 days! Moreover, as no atmospheric forcing is applied in R100, the aplied vertical mixing scheme is of secondary importance. We do not see, why the *boundary data transfer … seems incomplete*.

2. The simulations in the high-resolution nest are carried out without wind forcing, because with wind forcing, the surface variability in salinity is much weaker than the observed ocean color variability. All analysis is based on this simulation without wind forcing.

We have never stated that *with wind forcing, the surface variability in salinity is much weaker than the observed ocean color variability*

3. There is no model validation carried out at all, although high-quality observational data should be available, as described in lines 8-14.

Yes – there are observations available from "Expedition Clockwork Ocean", but the observations during that expedition where confined to rather small areas (less than a square kilometre), focusing on isolated submesoscale patterns. Hence, these observations are not suited for any validation.

The only qualitative comparison between model results and observations shows a snapshot of the simulated sea surface salinity (without wind forcing) and an ocean color image (representing chlorophyll). There is no evidence that those patterns should be similar.

Both salinity and chlorophyll are passive tracers which can be used as proxies for circulation patterns. Therefore, it is absolutely legitimate to compare them. Moreover, the R500 run **with** wind forcing was used.

Therefore, I recommend to reject this submission.
Some detailed comments to pages 1 – 8:
Page 1:
8: "high vertical velocities" instead of "strong vertical speeds" OK
11: What is the tendency equation?

The tendency equation is explained comprehensicely in 4.3.1. Experts on submesoscale dynamics should be familiar with it.

18: delete "ambitious". All research should be ambitious. OK
19: The word "turbulent" is missing here. OK
20: Here, turbulent kinetic energy seems to be used with a different meaning than usual. TKE is generally understood as non-hydrostatic which cannot be reproduced by a hydrostatic model. Do you here mean eddy kinetic energy (EKE)?

The Reviewer apparently associates TKE with turbulent kinetic energy occurring in vertical mixing processes – therefore *non-hydrostatic*. However, in the manuscript we never used the expression *turbulent kinetic energy* and consider only the *total kinetic energy* of horizontal processes.

22-24: Does this whole range of scales also occur at one location? I thought that sub-mesoscale turbulence is defined by non-dimensional parameters such as the Rossby number or the Richardson number (both being order of unity or larger). In the Bornholm Sea, you find probably not find submesoscale features of 100 m scale in the vertical.

Lines 22-24 are intended as a general introduction into the subject submesoscale turbulence. There is not at all a relationship with the Bornholm Basin. And the question *Does this whole range of scales also occur at one location?* reveals that the Reviewer has no clue what's is about.

Page 2:
1-5: I feel that this discussion of the many order of magnitude is an unnecessary overstatement. There is no turbulence on the 1000 km scale

This is not true! We are talking here about 2-dimensional turbulence; and the gyres in the ocean are elements of 2-dimensional turbulence (at the 1000-km scale, it is called geostrophic turbulence)

and I would base the definition on these non-dimensional numbers only. I propose that the Rossby number is defined here. For a Rossby number of order of unity, I would not say that Earth rotation is on minor importance. That should be the case only for much larger Rossby numbers.

That's a matter of opinion.

6-14: The red cascade is in contradiction to the spontaneous emission of subme-soscale structured from the meso-scale. So, we indeed have a (blue) forward cascade. The contradiction is due to the fact that there are about 40 years between the results referenced here. Therefore, I recommend to reformulate this paragraph.

The mesoscale feeds both the gyre scale (red cascade) and the submesoscale (blue cascade). This is not contradicting Charney (1971) or Rhines (1979). Those classical papers focused only on the feedback of (2-d) turbulent kinetic energy from the meso/synoptic scale to the larger scale.

Page 4:
16: Explain what an HBM operational model is.

Is explained in Section 2.2

Page 5:
10: It is a bit surprising that daily mean fields are used to forced a model simulations

which is designed to reproduce highly variable dynamics. The reasoning for this needs
to be motivated.

Daily fields are fully sufficient for the forcing of a model with 500-m resolution. In addition, the
boundary conditions are updated at every time step by interpolation in time and nudging.

18: It is unclear why different turbulence closure models have been used for the 500
m and 100 m models. And specifically confusing is that the large-scale closure model
KPP is used for the very high-resolution simulation of 100m. There will be many rel-
evant processes at that scale which are not reproduced by the KPP such as static
instabilities induced by differential advection. I would also suspect that the KPP model
does not reproduce the logarithmic bottom boundary layer.

Static instabilities in ROMS are removed by a convective adjustment algorithm; a quadratic law is
used for the botom friction. See P5L19 and our reply to item 1 above.

27: The use of the s-coordinate should be mentioned in the paragraph above, along
with the information about the number of vertical layers.

The use of s-coordinates is mentioned on P5L14, the number of layers is given in Table 1

I find it very unusual to base a high-performance model simulation on such a weak data basis. The
boundary data must include depth information, and with that the depth should be known.

But the reality is that the water depth is **not** included in the HBM output available at CMEMS!

Page 6:
30-32: This information about the turbulence closure model has partially been given
before (page 5). The additional information given here should be moved up to section
2.3.

OK

Page 7:
6-7: Length scales are partially given in metric units and partially in nautical miles. I
recommend to use metric units.

Generally, metric units are used. However, when graphics are based on a WGS84 coordinate system
nautical miles are better suitable because 1 arc minute in latitude = 1 mile.

8-9: Which gap is there between the meso-scale and the submeso-scale?

We mean the spectral gap in wavenumber space

12: TKE for Total Kinetic Energy is very confusing, also since turbulent kinetic energy
is used as well. I propose using KE for Total Kinetic Energy.

OK

Figure 3: It might not be clear to everyone what "cumulatively averaged" TKE is. It
should also be stated that this is the kinetic energy per unit mass.

Cumulative average is a well-known method in data analysis, indicating when a system attains stability (e.g. see the function *cumsum.m* in Matlab).

17-25: It does not become clear, why the unforced simulation is carried out and why it has been performed. Either do not discuss it or make sure that it gives a clear message. I specifically do not see how the spin-up time of the 500m nest can be estimated by this procedure.

This is extensively explained on P7L10-25

Lines 30-32 may contain an insight from this exercise.

Of course! More references of similar findings will be added there (thanks to the other Reviewer), which support our findings.

But it is not clear why vertical mixing should enhance horizontal mixing that would blur the fine structures. And if that happens, it should probably be a relevant process that needs to be discussed here in depth.

Atmospheric forcing does not simply induce vertical mixing. The wind forcing impacts the momentum transport on the scale of the forcing pattern, which in turn modifies the local vertical mixing and perhaps blocks the restratification by mixed-layer instability. A paper considering the impact of atmospheric forcing on the evolution of submesoscale patterns is in the works.

P 7, 33 – p 8, 8: It is not clear why one quantity in the model simulations (salinity) is validated with another quantity (ocean color). The latter is certainly related to chlorophyll which in turn might be related to phytoplankton concentration, a possibly positively buoyant particulate matter. It is not clear, why these two quantities should be related.

See above 3.

Why do you not compare observed surface temperature with simulated surface temperature?

The mesoscale distribution of SST can only derived from satellite IR measurements. However, the resolution of those is too low.

10-19: I have read this section many times, and I am quite sure that you say that you carry out the 100m simulation without any surface forcing because it would blur the submesoscale structures. It that really true?

Yes – it is true! More references will be added.

It is well acknowledged in the literature that submesosales are driven by surface forcing as well. So, if the model results detoriate due to the surface forcing, then the consequence should not be shut off the forcing but to find the reason between model results and simulations.

This was not the objective of this article.

Something seems to be fundamentally wrong.
If I see it right, a numerical is analysed here of which the only single validation is

the qualitative comparison between a snapshot of surface salinity and an ocean color snapshot. There is wind forcing in reality and no wind forcing in the model. Some of the structures look similar. That's all. Afterwards, the model results are intensively analysed in terms of submesoscale dynamics.

Apparently, the Reviewer stopped reading at P8.

Clearly, that is not science.

But this final remark is an offense to the authors! The first author (R. Onken) is visible in the oceanographic community since 1982, and he authored 40 articles in peer-reviewed journals and books. Frequently, Onken had disagreements with reviewers, but none of them criticised any of my manuscripts as "that is not science".
It is easy to make such derogative comments as an anonymous reviewer!

---

## Author Response (AR1)

Haine1998Haine1998Haine and Marshall(1998) Haney1991Haney(1991) Haidvogel2000Haidvogel et al.(2000) Hide1958Hide(19 Holton1982Holton(1982) Holton2004Holton(2004) Hoskins1982Hoskins(1982) Kantha1994Kantha and Clayson(1994) Klymak2016Klymak et al.(2016) Kubryakov2015Kubryakov and Stanichny(2015) Large1994Large et al.(1994)

– Version of 31 July 2019 –

[revised manuscript text omitted]
 2016. ~~The sea surface temperature of a limited area was first scanned with an infrared camera onboard an aerial motor glider because the location and time of the occurrence of STPPs is not predictable. As soon as a temperature signal of an STPP was localised, various measuring platforms were used for rapid repeat measurements of the three-dimensional structure and the temporal evolution of that feature before its decay within a few hours. Measurements were performed with research vessels, autonomous underwater vehicles, and even a 75-m long zeppelin. The latter "parked" over the STPP in order to record its life cycle with very high-resolution infrared and hyperspectral cameras. (http://www.uhrwerk-ozean.de, last access 20 September 2018).the~~ 
[revised manuscript text omitted]
 < 0$ in those parts is primarily supported both by $Q_{geo} < 0$ and $Q_{dv} < 0$. A bimodal structure is also visible for F4. Here, frontogenesis is supported both by $Q_{geo}$ and $Q_{ageo}$, and frontolysis only by $Q_w$. For F3, the situation is similar.

The above results conflict to those obtained by Capet et al. (2008b), referred to as CMMS in the following: according to their Fig. 7, the "residual" (equivalent to $F$ in this paper) at the sea surface is generally negative and the geostrophic contribution is always positive, while the corresponding quantities in the current article exhibit a clear bimodal structure. On the other hand, the contributions from ageostrophic straining and vertical diffusion are bimodal in CMMS but $Q_{ageo}$ and $Q_{dv}$ are predominantly positive. Similarly, the impact from vertical straining in CMMS is everywhere negative, in contrast to $Q_w$ that is bimodal as well

and the sign is alternating along-front. The cause for these disagreements might be the different horizontal resolution of ROMS (750 m in CMMS and 100 m in R100). By contrast, the multimodal structures and the alternating along-front sign changes of $F$ and its components resemble more those obtained by Gula et al. (2014): these are the bimodel (or even three-modal with filaments) structures and the alternating along-front sign changes.

5    Prominent structures of the tendency terms are also visible in C3 and C4, but those will be investigated in detail in a follow-up paper investigating the dynamics of a submesoscale cyclone.

**4.4.2  Submesoscale upwelling in eddies**

As noticed above, the vertical motion pattern in almost the entire model domain is impacted by internal waves that frequently blur the corresponding signals of STPPs. However, in some settings, the vertical motion related to submesoscale
10    fronts and eddies may supersede the internal wave-driven vertical velocity which can be seen in Fig. 7d. Such a situation is given during the life cycle of the submesoscale eddy C3 depicted in Fig. 11. C3 originates from a dense (cold and salty) streamer that invaded the area from the south, starting on 24 June around noon. While the streamer stretched farther to the north, it rolled up into C3 on 25 an 26 June (Fig. 12a–c; the entire process is shown in an animation on the web site https://www.hzg.de/institutes_platforms/coastal_research/operational_systems/coast_ocean_measurement/topics/index.php.de#tab-
15    100, last access 1 March 2019). In Fig. 12d–f, the vertical speed at 5-m depth is shown for the same period of time. On 24 June before the winding-up of the eddy started, the vertical motion pattern is controlled by a train of internal waves travelling from northeast to southwest. One day later, the impact of the eddy formation on the vertical motion field becomes visible: two extended patches with vertical speeds of either sign are located south and north of the eddy core. On 26 June, the pronounced downwelling area is smaller but the upwelling still prevails at the same location. Apparently, the eddy-driven vertical motion
20    supersedes the signal of the internal waves and causes persistent upwelling for a day or even more as can be seen in Fig. 12f: at the position of the crosshair, upwelling is still visible, although the eddy centre has moved already to the north by about 2 nm.

**4.4.3  Instability mechanisms**

The results above impressively show that submesoscale eddies grow rapidly in the R100 domain, preferably in the shallow areas with water depths less than 40 m (Fig. 7). However, it is not yet clear if inertial, symmetric or mixed-layer instability
25    drive the eddy growth. In the following, the R100 fields are analysed for criteria necessary for the occurrence of any type of these instabilities.

For inertial instability, negative absolute vorticity, $\zeta^a$, is  a necessary condition, which is equivalent to $\zeta^a = \zeta + f < 0$ or $\zeta/f < -1$. It can be seen in Fig. 7b (blue regions) that $\zeta/f < -1$ is satisfied along the coast of Bornholm and at a few isolated locations along the meridional open boundaries, i.e. close to the centre of the cyclonic eddy in the northwest corner and between
30    $54°25'$ and $54°30'$ N at the eastern boundary. However, the latter appearences are ignored because they are potentially due to false advection effects of relative vorticity discussed above. A zoomed image of $\zeta^a/f$ in the surface layer of R100 (Fig. 13) shows that negative values are found at 7 locations around Bornholm, indicating the birthplaces of disturbances driven by inertial instability. All of them are directly attached to the coast, but favourable conditions for the growth of the disturbances

seem to prevail only at sites no. 2 and 5 (yellow circles). In detail, on 26 June negative absolute vorticity is visible at no. 2, 3, 5, and 7. But within the subsequent 3 days, anticyclonic eddies have only developed at sites no. 2 and 5. It is common to these sites that the coast is on the right, relative to the current direction, and that the relative vorticity is anticyclonic due to the no-slip condition at the lateral solid boundaries. However, at sites no. 2 and 5, the curvature of the coastline is anticyclonic in contrast

5   to the other sites. Apparently, a solid boundary on the right and anticyclonic curvature of that boundary, are are additional necessary conditions for the formation of eddies driven by inertial instability. These conditions are also satisfied at site no. 7, but there is no indication for eddy growth. It is most likely prevented by the southward current with cyclonic vorticity along the west coast of the island. The above findings are in agreement with  Väli et al. (2017) who found values of $\zeta/f < -1$ at various near-coast locations in the Gulf of Finland. Gula et al. (2016b) showed that equivalent conditions were

10  satisfied in the Gulf Stream where the anticyclonic shear is amplified by the topographic drag against the slopes of the Great and Little Bahama Banks on its way through the Florida Straits. A similar situation is given where the California Undercurrent passes along Point Sur Ridge, a topographic obstacle near Monterey Bay (Dewar et al., 2015; Molemaker et al., 2015). Both cases lead to the formation of unstable submesoscale fronts and eddies.

According to Thomas et al. (2013), symmetric instability occurs when

$$\phi_{Ri_{geo}} < \phi_c \quad , \tag{10}$$

with

$$\phi_{Ri_{geo}} = \tan^{-1}\left(-\frac{1}{Ri_{geo}}\right) \tag{11}$$

and

$$\phi_c = \tan^{-1}\left(-\frac{\zeta_{geo}^a}{f}\right) \quad . \tag{12}$$

20  Here,

$$Ri_{geo} = \frac{f^2 N^2}{|\nabla_h b|^2} \tag{13}$$

is the Richardson number of the geostrophic flow,

$$b = -\frac{g\rho}{\rho_0} \tag{14}$$

is the buoyancy,

$$N^2 = -\frac{g}{\rho_0}\frac{\partial \rho}{\partial z} \tag{15}$$

the squared Brunt-Väisälä frequency, and

$$\zeta_{geo}^a = f + \frac{\partial v_{geo}}{\partial x} - \frac{\partial u_{geo}}{\partial y} \tag{16}$$

the absolute vorticity of the geostrophic flow $\boldsymbol{V}_{geo} = (u_{geo}, v_{geo})$. $\rho_0$ is a constant reference density, and $g$ the gravitational acceleration. Specifically, in regions where $\zeta^a_{geo}/f > 1$ (cyclonic vorticity), symmetric instability prevails, if the conditions

$$C_{SI} = -90° < \phi_{Ri_{geo}} < \phi_c \qquad \wedge \qquad \phi_c < -45° \qquad\qquad (17)$$

are satisfied. By contrast, symmetric instability is the dominant mode of instability in regions of anticyclonic vorticity ($\zeta^a_{geo}/f < 1$), if

$$C_{SI} = -90° < \phi_{Ri_{geo}} < -45° \qquad \wedge \qquad \phi_c > -45° \qquad . \qquad (18)$$

[revised manuscript text omitted]

~~Observations of STTPs covering a large area originate predominantly from spaceborne or airborne platforms, and they are mostly limited to measurements of the reflected and/or emitted radiance in the visible and infrared part of the spectrum. A spectacular satellite image of a phytoplankton bloom (Fig. ??, courtesy of NASA, National Aeronautics and Space Administration) exhibits the submesoscale spatial variability in the Baltic Proper. The shortest wavelength of meanders driven by hydrodynamical instability is around 1 km, the width of filaments is $\mathcal{O}(100)$ m, andthe diameters of the cyclonic eddies in the southwest and northeast are about 530 km, respectively. The scales in this image are similar to; the comparison is legitimate because the Rossby radius in this area is 5.7 km in summer (Fennel et al., 1991), which iscorresponding radii in the Arkona and Bornholm Basins (see Section 3). Note also the fringe-like features along several fronts with wavelengths and amplitudes < 100 m. As all fringe point to the same direction, it is unlikely that they are generated by any type of hydrodynamical instability. Moreover, it is supposed that they are created by the interaction of surface gravity waves and Langmuir cells with fronts (Nobuhiro Suzuki, personal communication, 2018). Similar fringe also become visible in the ocean fronts of ? (see Figure 2b there), when cross-front winds are directed from the less-dense side of the front to the denser side. Extremelyimages obtained from other aerial observations reveal that even smaller-scale eddies exist.~~ 
[revised manuscript text omitted]

| $rx_1$ | 1 | 1.75 | 0.84 | stability condition after **?** |

[Figure]

**Figure 1.** The western Baltic Sea. Water depth is given in metres. The approximate experimental area of the "Expedition Clockwork Ocean" is indicated by the red polygon.

[Figure]

**Figure 2.** Top-layer salinity in HBM (left column), R500 (centre), and R500_NF (right) for 1, 10, 20, and 30 June. The HBM fields are interpolated onto the R500 horizontal grid. Due to the coarser horizontal grid of HBM, inland lakes are not resolved and salinity values were assigned to places that are dry in R500.

[Figure]

**Figure 3.** Instantaneous ($KE_{R500}$, $KE_{R500\_NF}$) and cumulatively averaged ($\overline{KE_{R500\_NF}}$) kinetic energy per unit mass of the model runs R500 and R500_NF. The wind speed is shown as black curve. All quantities are averaged over the model domain.

[Figure]

**Figure 4.** (a) Top-layer salinity of the R500 model on 23 June 09:00. (b) RGB composite from the Ocean and Land Colour Instrument of ESA satellite Sentinel-3 for the R500 domain on 23 June 09:32. The original image was manually adjusted in order to fit to the Mercator projection used in (a). C1, C2, and AC refer to the signatures of cyclonic and anticyclonis eddies, respectively. For the Oderbank, cf. Fig. 1.

[Figure]

**Figure 5.** R100: top-layer salinity (left column), potential temperature (centre), and the absolute horizontal gradient of potential density (right) on 20, 23, 26 and 29 June. The upper end of the colour axis of $|\nabla\rho|$ is limited to $4\times10^{-4}$ kg m$^{-4}$ to distinguish higher gradients from the blue background.

[Figure]

**Figure 6.** R100: frequency distribution of $|\nabla\rho|$ in the top layer. The initial and the open boundary conditions for R100 were provided by (a) R500 and (b) R500_NF, respectively. The values were binned in 10 intervals of $0.5 \times 10^{-3}$ kg m$^{-4}$ width between 0 and $5 \times 10^{-3}$ kg m$^{-4}$. Frequencies < 100 representing less than 0.1% of the total number of horizontal grid points were not considered as significant.

[Figure]

**Figure 7.** R100, 26 June: (a) Magnitude and direction of top-layer horizontal velocity $\boldsymbol{V}$ (vectors are plotted at 3-km resolution), (b) relative vorticity $\zeta$ and (c) horizontal divergence $\delta$, each scaled by $f$, (d) vertical velocity $w$ at 5-m depth. The white line in (a) is the 40-m depth contour. The red box in (a) refers to the zoomed area shown in Figs. 11 and 12, the box in (c) is the zoomed area of Fig. 9, and the box in (d) is the zoom shown in Fig. 15.

[Figure]

**Figure 8.** R100 with full atmospheric forcing on 26 June: (a) Magnitude and direction of top-layer horizontal velocity $\boldsymbol{V}$ (vectors are plotted at 3-km resolution), (b) relative vorticity $\zeta$ and (c) horizontal divergence $\delta$, each scaled by $f$, (d) vertical velocity $w$ at 5-m depth. The white line in (a) is the 40-m depth contour. For all subplots, the same color scaling was used as in Fig. 7.

[Figure]

**Figure 9.** R100, 26 June: zoom of $f$-scaled horizontal divergence $\delta$ (grayscale image, for the position of the zoomed area see Fig. 7c). Bright areas indicate divergent flow, convergences appear dark. The magenta lines show the $|\nabla\rho| = 10^{-4}$ kg m$^{-4}$ contours. The yellow arrows likely mark internal wave packages as discussed in the text. The horizontal yellow line is the position of the section shown in Fig. 10. The black/yellow-dashed ruler represents a distance of 5 km

[Figure]

**Figure 10.** R100, 26 June: zonal section of vertical velocity $w$ at 54°49.9' N. Green contours indicate potential density anomaly $\sigma_\theta$, bold solid and dotted contours are isotachs of the meridonal velocity component $v$ [cm s$^{-1}$]. The position of the section is marked in Fig. 9.

[Figure]

**Figure 11.** R100: near-surface properties on 26 June within the red box depicted in Fig. 7a. (a) Absolute horizontal density gradient $|\nabla\rho|$ [$10^{-4}$ kg m$^{-4}$], and (b) scaled relative vorticity $\zeta/f$ in the top layer, (c) vertical velocity $w$ [m day$^{-1}$] at the base of the top layer. (d) – (i) The frontal tendency $F$ and the components of the tendency equation $Q_{geo}$, $Q_{ageo}$, $Q_w$, $Q_{dv}$, $Q_{dh}$ [$10^{-13}$ kg$^{-2}$ m$^{-8}$ s$^{-1}$] at 2-m depth. Red marks frontogenesis, blue frontolysis. For the labels F1–F4, C3, C4 see text. The black/green-dashed ruler represents a distance of 2 km, vectors in (e) and (f) are drawn at 600-m intervals.

[Figure]

**Figure 12.** R100: (a)–(c) temporal evolution of potential density anomaly $\sigma_\theta$ in the top layer with vectors of total velocity $\boldsymbol{V}$ drawn at 400-m resolution, (d)–(f) corresponding vertical velocity $w$ at 5-m depth; upwelling is positive. The area shown is indicated by the red box in Fig. 7a. The crosshair tags the same position in all subplots. The black/red-dashed ruler represents a distance of 2 km.

[Figure]

**Figure 13.** R100: normalized absolute vorticity, $\zeta^a/f$, in the surface layer of the waters around Bornholm on 26 and 29 June. Encircled numbers indicate near-coastal locations of $\zeta^a/f < 0$. Yellow circles mark locations where inertial instabilities tend to grow offshore.

[Figure]

**Figure 14.** R100: (a, b) $\phi_{Ri}$, (c, d) $\phi_c$, and (e, f) $C_{SI}$ at 3-m depth on 15 (left column) and 26 June (right). (e) and (f) are logical maps indicating where the condition for symmetric instability, $C_{SI}$, is satisfied in regions of cyclonic (red) and anticyclonic (blue) absolute vorticity, according to eqs. (17) and (18), respectively. For explanation of the other symbols see text.

[Figure]

**Figure 15.** R100, 26 June: Potential density anomaly $\sigma_\theta$ in the top layer with total velocity **V** drawn at 600-m resolution. For the position of the area, see the red box in Fig. 7d. The black box indicates the position of the maps shown in Fig. 16a–e. The black/white-dashed ruler represents a distance of 2 km.

[Figure]

**Figure 16.** R100, 26 June: maps of dynamical quantities within the black box in Fig. 15. (a) $\sigma_\theta$ and $\boldsymbol{V}$, (b) $|\nabla\rho|$ and $\boldsymbol{V}$, (c) $\zeta/f$ and $\boldsymbol{V}_{geo}$, (d) $\delta/f$ and $\boldsymbol{V}_{ageo}$ in the top layer, and (e) $w$ and $\boldsymbol{V}_{ageo}$ at the bottom of the top layer. Velocity vectors are drawn at a resolution of 200 m. Vertical sections of (f) $v$, (g) $u_{ageo}$, (h) $\zeta/f$, (i) $\delta/f$, and (j) $w$. Contour lines of potential density anomaly $\sigma_\theta$ are green and spaced at the same intervals as in (a). The horizontal dashed lines in the centres of (a)–(e) indicate the position of the zonal sections displayed in (f)–(j). In either subplot, the red/yellow-dashed rulers represent a horizontal distance of 500 m.

[Figure]

**Figure 17.** R100, 26 June: Richardson number $Ri$ at 2-m depth and total top-layer velocity $\mathbf{V}$ drawn at 600-m resolution. For the position of the area, see the red box in Fig. 7d. The magenta box indicates the position of the maps shown in Fig. 16a–e. It is identical with the black box in Fig. 15. The black/white-dashed ruler represents a distance of 2 km.

[Figure]

**Figure 18.** Passive tracer patterns of submesoscale eddies. (a) Observed sea surface temperature of a cyclonic eddy in the Southern California Counter Current. The image was taken on 1 February 2013 at 20:34 in the framework of the SubEx experiment (Marmorino et al., 2018). (b) Modelled salinity and (c) potential temperature of the cyclone C3 on 26 June at 15-m depth (cf. Figs. 11 and 12). Cold water spots in (a) are denoted by "S". The black/white-dashed ruler in (b) and (c) represents a distance of 1 km.

---

## Referee Report (RR1)

I am satisfied with the authors' responses to my criticism and the corrections they made to the text of the article. The revised article can be published as is.

---

## Author Response (AR3)

Dear Reviewer#3,

Thank you very much for your third review of our manuscript. Please find below our replies to your comments. Note that your comments are written in blue while our replies are black.
* * *
**Reply to your comments of 02 January 2020**
* * *
Onken et al. Very high-resolution modelling of submesoscale turbulent patterns
and processes in the Baltic Sea; Ocean Sci. Discuss., https://doi.org/10.5194/os-2019-44

Review 3

Final comments

**(1)** I am not satisfied with the response of the authors to some of my previous comments and questions. The major problems are related to the authors' concept that the internal waves are part of submesoscale processes

Action: see below **(5)**

**(2)** and the set-up of models – too low number of vertical layers,

In our previous reply to your comments of 20 September 2019, we have justified the number of vertical levels at length in item **(20)**. You may note as well that for technical reasons the number of sigma-layers in a ROMS-to-ROMS nesting must not change. Hence, the number of layers in R100 is the same as in R500.

Action: none

**(3)** nesting of models with and without atmospheric forcing,

This issue was already discussed in detail in our previous reply to your comments of 18 July 2019. We are surprised that you rake it up again as you seemed to be satisfied according to your comments of 20 September.

Action: none

**(4)** different turbulent parametrization, etc.

As far as we can see, the different parameterisations were never criticized by you – nor by your comments of 18 July, nor by those of 20 September.

Action: none

**(5)** The authors refer to a paper by McWilliams (2016) to support their approach that internal waves are also among the submesoscale turbulent patterns and processes. At the same time, Williams (2016) clearly distinguishes between the internal gravity waves (IGW) and submesoscale currents (SMC). The scales of SMC and IGW are coinciding, but they are two separate branches of the flow of energy from mesoscale processes to microscale dissipation. Although in the term "submesoscale processes", the scale is mentioned, it is more than that. See, e.g., a definition by McWilliams (2019): "Besides their identifying scales, submesoscale currents are distinctive in their flow

patterns, their essential dynamical processes, and their consequences for transport, mixing, and dissipation in the general circulation". I think it is misleading to define internal waves and submesoscale processes as submesoscale turbulent patterns and processes. It also causes a misleading statement that submesoscale processes are characterized by strong vertical speeds of O(100 m day-1).

OK – we agree: after a thorough rereading of McWilliams (2016), we found in the Introduction "This SMC scale range overlaps to a high degree with IGWs, and the two phenomena must be distinguished by their evolutionary behaviors,..."

Action:
- The corresponding piece of text in the Introduction was rephrased (new ms P2L25-26)
- A paragraph in the Conclusions was rewritten (P22L3-9)
- *O(100) m day$^{-1}$* was substituted by *O(10) m day$^{-1}$* in the Abstract (P1L10)

**(6)** Since some questionable assumptions were used and clearly different settings were applied to the nested models, readers still might wonder what results are of general value and what results are related to the approach. For instance, the authors state that the submesoscale features appear preferably in the shallow areas with water depths less than 40 m. Is it a natural phenomenon or is it related to the model resolution? One could argue that in the deeper areas, the vertical resolution of the model is too coarse to simulate submesoscale processes.

According to McWilliams (2019), the principal SMC generation mechanisms are (1) extraction of available potential energy in the weakly stratified surface layer ("mixed-layer instability"), and (2) topographic-drag vorticity generation in flows along a sloping bottom. As strong bottom slopes are found only along Rönnebank and in the south of the R100 domain (see Fig. 1), the mechanism (2) may be significant in those areas. However, for water depths > 50 m, the bottom is mostly flat and topographic-drag vorticity generation can be excluded. Hence, mixed-layer instability is the only generation mechanism there, and that does not require higher vertical resolution.

Action:
- The enhanced submesoscale activity in shallow areas is discussed (P12L29-P13L2)
- McWilliams (2019) was added to the References

**(7)** There are also smaller issues, which could be omitted to allow to shorten and focus the manuscript better. For instance, it is not relevant to discuss the long-term precipitation and evaporation to justify why salinity is an ideal parameter (passive tracer), etc.

Remember that **you** asked both in your reviews of 18 July (item **(9)**) and 20 September (item **(8)**) for an explanation why salinity is the ideal parameter.

Action: The corresponding discussion was slightly shortened. See P7L15-17.

On the other hand, the authors have conducted a thorough analysis of their results using a series of parameters and their distributions related to certain processes, patterns or mechanisms and compared the findings with other papers/authors. Thus, I think the paper is worth to be published to intensify the discussions on the nature and features of submesoscale processes. I leave the decision to the Editor(s).

McWilliams J.C. 2016. Submesoscale currents in the ocean. Proc. R. Soc. A 472: 20160117. https://doi.org/10.1098/rspa.2016.0117.

McWilliams J.C. 2019. A survey of submesoscale currents. Geosci. Lett. 6:3.
https://doi.org/10.1186/s40562-019-0133-3.

Dear Reviewer#4,

Thank you very much for your review of our manuscript. Please find below our replies to your comments. Note that your comments are written in blue while our replies are black. P(age) and L(ine) numers refer to the new manuscript (ms) except for if otherwise stated.

Our replies below are ordered alphabetically **(A), (B), (C)**, ... in contrast to our replies to your comments of 7 October which were ordered **(1), (2), (3)**
* * *
**Reply to your comments of 27 November 2019**
* * *
Referee report of the re-revised manuscript "Very high-resolution modelling of submesoscale turbulent patterns and processes in the Baltic Sea" (version of 12 Nov 2019) by Reiner Onken, Burkard Baschek, and Ingrid M. Angel-Benavides, submitted for consideration to Ocean Science.

I would like to thank the authors for their reply and the changes implemented to the manuscript. Unfortunately, some points in their reply have left me uneasy. My most major concern about this manuscript was and still is the modelling methodology. I was surprised by some of their choices and asked the authors to better justify them or explain the process which lead to those choices. For some issues they did so, for which I thank them. But for other chose not to do so, and for some issues I did not find their justifications convincing.

As sound methodology is one of the most important aspects of any scientific study, it is of utmost importance that modellers do their due diligence when designing modelling experiments. I am not yet convinced this has been the case in this study. Therefore I have no choice but to recommend another major revision. I hope the authors would take some time to think what can be considered a "reasonable effort" on their part to make sure they can justify their choices adequately, and whether they feel confident that their modelling configuration is well enough documented in the manuscript, along with any potential weaknesses. I emphasize that I do not claim that there necessarily are any fundamental flaws in the methodology, but rather that their replies (and in some cases the lack thereof) have left me with insufficient information to determine if this is so.

Please find my specific comments below. Numerals in parenthesis refer to the replies made by the authors in response to my original review. At this time I refrain from making what I would consider minor comments so they do not shadow these more major issues.

**(A)** In their reply, the authors do not address in any way my three main comments presented under "General comments". I was truly surprised by this decision. These were the questions I was most looking to be answered, but there are no answers. Because of this I feel I lack information to make a fully informed review. Their other replies do not really address these comments fully.

Please find our replies below.

**(B)** Regarding my first major comment asking the paper's objectives to be clarified, I see that there have been some edits to the introduction. On one hand I feel this comment could have been addressed more carefully. However, if the authors find their manuscript now accessible enough and do not want to address this in more depth, I have no further comments on this issue.

Action: The corresponding paragraph in the Introduction was rewritten: P2L1-8.

**(C)** Regarding my second and third point about documenting the limitations of the study and

justifying their methodology, I see improvement and that needs to be acknowledged. However, I feel it would have been possible for the authors to be much clearer on this issue.

Concerning the limitations of the study, we have added or modified pieces of text in the Conclusions and in the Abstract.

Action: see P21L30-P22L2 and P1L6-9

**(D)** For example, regarding their reply (8), I don't think that "worrying the readers" is an acceptable reason to omit information that would help the readers make their own conclusions about the study.

Action: According to your desire in the previous review, we have expanded the description of the CMEMS product in Section 2.2, see P5L8-16.

**(E)** This approach of omitting information is deeply worrying and leaves me to wonder what else has been omitted from the manuscript because the readers would find it upsetting.

This remark is speculative, polemic, and provocative.

Action: none

**(F)** Also regarding reply (8), I would again ask the authors to insert into the text the version number of the CMEMS product they used. This version number is identified on the cover of the product user manual (and in the following change record) as e.g. "4.0" or "3.0". The issue of the user manual is useful information, but it is not the same thing as the version of the product.

At the time when we started the modeling activities in 2017, the PUM version was 2.1.

Action: see P5L12

**(G)** In their reply (9), the authors indicate they have not really done any kind of survey of available boundary condition products before settling on the CMEMS product, or indeed even considered other options. They also say they "cannot remember" if the reanalysis product was available at that time. In my opinion, "due diligence" or "reasonable care" would have required at least rudimentary survey of available state-of-the-art options for boundary data, along with their main features. I sincerely hope this reply was some kind of misunderstanding, perhaps due to linguistic challenges.

No - it was **not** some kind of misunderstanding. We checked again the Product User Manual of BALTICSEA_REANALYSIS_PHY_003_011 and found on P5/16 that the product was released in April 2018, hence it was not yet available when we started our modeling activities in early 2017. The coarse horizontal resolution of 4 km would also have required a triple-nesting setup (4 km --> 800 m --> 267 m --> 89 m) in order to arrive at a resolution of 100 m because a nesting ratio of 5 should not be exceeded, and the ratio should be uneven (i.e. 5 or 3) for the ROMS-to-ROMS nesting. To the best of our knowledge, there was no alternative to CMEMS-HBM in 2017, and a *rudimentary survey of available state-of-the-art options for boundary data, along with their main features* was not required.

Action: A few remarks concerning the choice of CMEMS-HBM have been added (P5L21-23)

**(H)** In their reply (10) they state that they did not do any effort to obtain the bathymetric data of their boundary data. In fact, they still have not done so after being asked about this by myself and earlier by another referee. Rather, in their reply they just state their assumption on what this file

"probably" is. Yet they still cite the lack of this file as a rationale for their nesting procedures in the manuscript. In my opinion, in this case a reasonable requirement would have been at least one email asking for this file from the service desk. If they do not provide it after asking, fine. If the file is not useful for whatever reason, then that's fine too. But I do not understand why no effort was taken to obtain it, especially after the comments made by one of the other reviewers.

We have investigated again the issue with the bathymetry data and found that *the file would not have been useful for whatever reason*. For justification, we explain now step by step the downscaling procedure, starting from CMEMS-HBM where all prognostic variables are vertically interpolated from the HBM native grid onto standard depth levels:

(1) The CMEMS-HBM fields are interpolated horizontally on the higher-resolution R500 grid, but still on the same standard depth levels.

(2) All fields (except for the sea surface height) are interpolated vertically on the ROMS terrain-following vertical coordinates, the depth of which is determined *solely* by the ROMS-bathymetry. This bathymetry was already iteratively smoothed until the stability condition of Haidvogel et al. (2000) was reached everywhere.

(3) The vertical interpolation of the horizontal velocities $u$ and $v$ is not mass conserving, if CMEMS-HBM provides non-zero normal velocities close to topographic obstacles which are present in ROMS but not in HBM. If these velocities hit, for instance, a seamount, the the continuity equation in ROMS would create unrealistic vertical velocities.

(4) Such an issue occurred once in R500 at the northern domain boundary. There, the meridional velocity flew against a small abyssal hill that was apparently not existent in HBM. To cure that problem, the hill was flattened manually until the vertical velocities diminished.

As you can see, we could not have made use of the HBM bathymetry.

Action:
• The critical sentence in the old ms (P5L22-23) was removed.
• The text in bullet 4 was modified (P6L17-20).

**(I)** If we take replies (8), (9) and (10) together, they leave the impression that the authors have decided to use the CMEMS product as their boundary data without much consideration for its features, and now do not want to explain their choice or the main features of the dataset in the paper. This impression may very well be unintended, but I would kindly suggest to the authors to reconsider if it would in fact be better to just briefly explain these things to the readers to avoid such misunderstandings.

Action: see above **(D), (E), (F), (G), (H)**

**(J)** Based on my experience from the ocean modelling community, it is not accurate at all to call these model configurations "expensive in terms of computer resources" in the year 2019. Their reply (13) justifies this by claiming that this is a matter of opinion, but frankly, the given information seems to indicate that this model is not computationally expensive when compared to other commonly used configuration. This can be e.g. regional CMEMS configurations, or recently published coastal modelling configurations from the Baltic Sea basin. It is also run on a relatively modest computational system. The test phase of 3 months seems very much normal for this kind of a study, not exceptionally long at all. I ask the authors to remove this statement from the manuscript.

Action: The statement was removed. See P9L14.

**Very high-resolution modelling of submesoscale turbulent patterns and processes in the Baltic Sea**

Reiner Onken[1], Burkard Baschek[1], and Ingrid M. Angel-Benavides[1]

[1]Helmholtz-Zentrum Geesthacht, Max-Planck-Straße 1, 21502 Geesthacht, Germany

*Correspondence to:* Reiner.Onken@hzg.de

**Abstract.** In order to simulate submesoscale turbulent patterns and processes (STPPs) and to analyse their properties and dynamics, the Regional Ocean Modeling System (ROMS) was run for June 2016 in a subregion of the Baltic Sea. To create a realistic mesoscale environment, ROMS with 500-m horizontal resolution (referred to as R500) is one-way nested into an existing operational model, and STPPs with horizontal scales < 1 km are resolved with a second nest of 100-m resolution (R100). Both nests use 10 terrain-following layers in the vertical. The comparison of the R500 results with a satellite image shows fair agreement. While R500 is driven by realistic air-sea fluxes, the atmospheric forcing is turned off in R100 because it prevents the generation of STPPs and blurs submesoscale structures. Therefore, R100 provides a deep insight into ageostrophic processes and associated quantities under quasi-adiabatic conditions that are approximately met in no-wind or light-wind situations. The validity of the results is furthermore limited to the selected region and the time of the year. STPPs evolve rapidly within a about a day. They are characterised by  vertical speeds of $\mathcal{O}(\cancel{100}10)$ m day$^{-1}$ and relative vorticities and divergences reaching multiple of the Coriolis parameter. Typical elements of the secondary circulation of two-dimensional strain-induced frontogenesis are identified at an exemplary front in shallow water, and details of the ageostrophic flow field are revealed. The conditions for inertial and symmetric instability are evaluated for the whole domain, and the components of the tendency equation are computed in a subregion. While anticyclonic eddies are generated solely along coasts, cyclonic eddies are rolled-up streamers and found in the entire domain. A special feature of the cyclones is their ability to absorb internal waves and to sustain patches of continous upwelling for several days favouring plankton growth. The kinematic properties show good agreement with observations, while some observed details within a small cyclonic eddy are only partly reproduced, most likely due to a lack of horizontal resolution or non-hydrostatic effects.

**1 Introduction**

This article was motivated by the "Expedition Clockwork Ocean" which was conducted 20–28 June 2016 in the  Baltic Sea to the south of the island of Bornholm. The objective of that survey was to observe submesoscale turbulent patterns and processes (STPPs) in order to better understand their role in the cascade of turbulent kinetic energy in the ocean, and to assess their impact on the primary production.

Presently, the knowledge about STPPs in the corresponding area is primarily limited to eddy statistics and originates solely from space-borne remote sensing obervations (Gurova, 2012; Tavri et al., 2016; ?) and the model study of Vortmeyer-Kley et al. (2019). Some more information about kinematic properties of STPPs is conveyed by the high-resolution numerical study of Zhurbas et al. (2019), who showed that the overwhelming dominance of cyclonic eddies on satellite images is related to their higher angular velocity,

5  In the following, high-resolution modeling is used to  generate STTPs and to further improve our knowledge about their characteristics (i.e. tracer patterns, kinematics, impact of atmospheric forcing, fronts, instabilities, and eddies), in the corresponding region at the respective time .

[revised manuscript text omitted]
 operational model, the output of which is provided by CMEMS (Copernicus Marine Environment Monitoring Service, see below). An even finer nested ROMS model with a grid size of 100 m (R100) is used to enable the generation of STPPs, thus providing a base for an analysis of their properties and dynamics (Fig. 1).

The utilised numerical models are described in Section 2. The results of the numerical experiments are presented in Sections

25  3 and 4 and compared with observations in Section 5, followed by the conclusions. In the following, all time specifications refer to the year 2016 (unless stated otherwise) and are given in UTC (Universal Time Coordinated).

**2  The models**

**2.1  Geographic and oceanographic setting**

The Baltic Sea is a semi-enclosed marginal shelf sea with a mean water depth of 55 m and with narrow shallow connections to

30  the North Sea. Due to river runoff, the water balance is positive driving an estuarine circulation with quasi-permanent outflow of fresh surface waters and an intermittent inflow of salty water from the North Sea. At the surface, this creates a horizontal salinity gradient with high salinities in the west and almost freshwater conditions in the far north. Salinity is increasing with

depth thus stratifying the water column year-round and generating a permanent halocline at about 60-m depth in the deeper basins. For more details see Feistel et al. (2008), Leppäranta and Myrberg (2009), or Osiński et al. (2010).

The area of this model study is separated into two basins, the Arkona Basin and the Bornholm Basin (Fig. 1). The Arkona Basin is the smaller one with a maximum water depth of 51 m, while the maximum depth of the Bornholm Basin is 92 m. The

5 basins are connected by two channels, with an exchange of water limited by sills of 45 m depth in the Bornholmsgat (Magaard and Rheinheimer, 1974) and 31 m between Rönnebank and the island of Rügen.

**2.2 The CMEMS product**

The CMEMS product is provided at http://marine.copernicus.eu, product BALTICSEA_ANALYSIS_FORECAST_PHY_003_006, last access 17 Februar 2020) since April  2015. The product is based

10 on HBM (HIROMB-BOOS model (Berg, 2012)), which is an operational ocean circulation model predicting the physical conditions of the Baltic Sea. It is referred to as CMEMS-HBM in the following. CMEMS-HBM used in this study is documented in the Product User Manual CMEMS-BAL-PUM-003-006.pdf, Version 2.1, and the validation framework is described in the Quality Information Document CMEMS-BAL-QUID-003-006.pdf. HBM is running twice daily at DMI (Danish Meteorological Institute) in Denmark, and a backup production system is running at BSH (Bundesamt für Seeschifffahrt und Hydrography) in

15 Germany. While the DMI setup uses meteorological data from DMI-HIRLAM with a horizontal resolution of 5 km, the BSH version is driven by data from the Cosmo-EU model with 7-km horizontal resolution provided by the German Weather Service, DWD. CMEMS-HBM comprises daily mean and hourly instantaneous fields at a horizontal resolution of 1 nmi (nautical mile) at 25 depth levels spaced at 5 m between the sea surface and 100-m depth, and additional levels below at 150, 200, 300, and 400-m depth. For this article, the daily mean fields of June 2016 were utilised that contained 30 records of the prognostic vari-

20 ables potential temperature, salinity, horizontal velocity, and sea surface height for each June day.

The modeling activity for this study started in early 2017. At that time, the CMEMS-HBM reanalysis product, BALTICSEA_RENALYSIS_PHY_003_011 (also available at http://marine.copernicus.eu), was not yet available since it was

25 released for the first time in April 2018. Hence, there was no alternative or CMEMS-HBM at the required horizontal resolution.

**2.3 ROMS**

The employed numerical ocean circulation model ROMS is a hydrostatic, free-surface, primitive equations model. Its algorithms are described in detail in Shchepetkin and McWilliams (2005). In the vertical, the primitive equations are discretised over a variable topography using stretched terrain-following coordinates, so-called s-coordinates (Song and Haidvogel, 1994).

30 In the horizontal, spherical coordinates are used. Biharmonic mixing along isopycnic surfaces is applied to the tracers, both in R500 and R100, while biharmonic mixing of momentum is used in R500 and a monoharmonic formulation in R100. The vertical mixing of momentum and tracers is parameterised with the GLS (Generic Length Scale) scheme by Umlauf and Burchard (2003) in R500, and with the interior closure by Large et al. (1994) in R100, referred to as the KPP scheme. For the bottom

friction, a quadratic law is applied, and the pressure gradient term is computed using the standard density Jacobian algorithm by Shchepetkin and Williams (2001). The air-sea interaction boundary layer in ROMS is formulated by means of the bulk parameterisation by Fairall et al. (1996). R500 and R100 have the parameters and equations listed in Table 1 in common, while the individual grid-size-dependent parameters and properties are summarised in Table 2. As the spatial scales of the smallest

5  known STPPs are $\mathcal{O}(10\,\mathrm{m})$, it is expected that large and medium-scale STPPs are resolved.

**2.4 Nesting, boundary conditions**

There are two offline nesting steps: R500 is nested in CMEMS-HBM, and R100 is nested in R500. While the ROMS-to-ROMS nesting is technically straightforward, the first nest is somewhat more delicate, because the CMEMS-HBM output is provided on depth levels while ROMS uses s-coordinates.

10   Therefore, the setup of the R500 domain and the nesting was accomplished as follows:

1. The bathymetry of the GEBCO_2014 grid (General Bathymetric Chart of the Oceans, 30 arc seconds horizontal resolution) was used as the lower boundary of the R500 domain, and it was smoothed iteratively until the stability condition $rx_0 \le 0.2$ was reached everywhere (see Haidvogel et al. (2000) and Table 2).

2. The CMEMS-HBM prognostic variables were interpolated linearly onto the R500 horizontal grid at each CMEMS-HBM

15      depth level in 24-hour intervals and for each day of June.

3. The R500 vertical grid was defined according to Table 1 and the CMEMS-HBM fields were interpolated vertically onto the R500 vertical grid. The first record of the resulting data set served as initial condition for the R500 integration, while the lateral boundary conditions were extracted from all records.

4. R500 was integrated for one day, and the near bottom velocities were checked for odd features that might have been

20      caused by  non-zero normal velocities at topographic obstacles which were not resolved in CMEMS-HBM. If such features (e.g. abnormal vertical motions) were noticed, the R500 bathymetry was manually adjusted and the above procedure was iteratively repeated, starting with step 3.

For the R500-to-R100 nesting, the same vertical grid definition was used, and no interpolation from depth levels to s-

25  coordinates was required. Special care was taken for the preparation of the initial and boundary conditions, as a nesting ratio of 5 is rather challenging: cubic splines were used for the horizontal interpolation of the prognostic variables of R500 onto the child's grid, because the structures of jet flows across the open boundaries of the R100 domain looked more realistic than those obtained by linear interpolation. In addition, the downscaled fields were generated in 3-hour intervals, leading to a smoother temporal change of the lateral boundary conditions. Cubic splines were used as well for the interpolation of the atmospheric

30  forcing fields on the R100 grid. Because in a nested configuration the s-coordinates of the parent and the child at any location are only identical if the bathymetry is the same, the bathymetry of R100 was cloned from R500 and linearly interpolated onto the finer grid.

For each nest, radiation boundary conditions with nudging (Marchesiello et al., 2001) were applied to temperature and salinity, barotropic and baroclinic momentum, and the mixing of turbulent kinetic energy along the lateral boundaries. The boundary conditions of the free surface were defined according to Chapman (1985). In all ROMS setups, the nudging time scales were set to 2 days for the corresponding variables. At the surface, the air-sea interaction in the ROMS nests was specified by means of atmospheric forcing fields from the so-called "assimilation runs" of the ICON-EU model, provided by the German Weather Service (DWD). The output of these runs was considered to be the best available product, because the runs were initialised in 3-hour intervals at 00 h, 03 h, 06 h, . . . and driven by the most recent near-real time assimilation fields, in contrast to the forecast runs initialised semidaily at 00 and 12. The horizontal resolution of ICON-EU is about 6.5 km and output is produced in 1-hour intervals.

**3   R500: model results and validation**

STPPs are generated in the straining field of mesoscale eddies. According to Osiński et al. (2010), the condition for eddy resolving models of the southern Baltic is that the Rossby radius is resolved by at least 4–5 horizontal nodes. As the Rossby radii in the Bornolm and Arkona Basins are in summer around 7.2 and 3.7 km, respectively (Fennel et al., 1991), and since the grid size of CMEMS-HBM is 1 nmi, it is definitely not eddy-resolving or even eddy-permitting (2–3 nodes) in the Arkona Basin and perhaps eddy-permitting at best in the Bornholm Basin. The eddy-resolving R500 was initialised from CMEMS-HBM on 1 June 00:00 and integrated for 30 days. The vertical mixing in R500 is accomplished with GLS using the $k - \omega$ setup of Wilcox (1988) with the stability function of Kantha and Clayson (1994).

For a Lagrangian water parcel, the freshwater budget is controlled by $P - E$ which is the difference between precipitation $P$ and evaporation $E$. On the longterm average, $P - E$ is around  0.1 mm day$^{-1}$  (Smedman et al., 2005) that may cause maximum salinity changes on the order of $10^{-2}$ per month for typical mixed-layer depths of 10 m (see Section 4.4.3). In contrast, the heat budget is dominated by the short wave radiation flux leading to warming around $5°$ C of the near surface layers in June. Hence, salinity is the ideal parameter to trace turbulent patterns as in the Baltic Sea it behaves like a passive and quasi-conservative tracer. Fig. 2 shows salinity in the uppermost layer of CMEMS-HBM and R500 on 1, 10, 20, and 30 June (left and centre panels). R500 rapidly develops turbulent structures that are only marginally identifiable in CMEMS-HBM. These are, for example, the low-salinity outbreaks along the northern boundary, a high-salinity eddy in the Arkona Basin, and mushroom-like patterns east and southeast of Bornholm on 10 and 20 June. The horizontal scales of these features are $\mathcal{O}(10$ nmi$)$, but also smaller patterns with a horizontal extent of 5 nmi, or even less, are generated by R500, such as the filaments around Bornholm and the meanders immediately off the Polish coast on 20 June. Hence, R500 apparently bridges the gap between the mesoscale and the submesoscale.

R500 provides the initial and boundary conditions for R100. Insofar, it is worth knowing to what extent its generated two-dimensional turbulence is in a state of statistical equilibrium, and at what time during the integration an equilibrium state is attained. To determine this so-called spin-up time, the blue dash-dotted graph in Fig. 3 shows a time series of the domainavageraged kinetic energy per unit mass, $KE_{R500}$; it fluctuates strongly between $3 \times 10^{-3}$ and more than $12 \times 10^{-3}$ m$^2$ s$^{-2}$ and does not reach a stable value. Evidently, it is difficult to determine the spin-up time from R500 because $KE_{R500}$ is strongly impacted by wind bursts as shown by the black curve. Therefore, R500 was compared to a run where the atmospheric forcing was completely turned off by setting the air-sea fluxes of net heat, fresh water and momentum to zero. This run is referred

5    to as R500_NF ("No Forcing"). Here, the intense fluctuations of *KE* vanished as shown by $KE_{R500\_NF}$ (blue solid graph), but there are still some smaller-scale oscillations with maxima on 11, 14, 20, and 28 June, the existence of which impede the estimate of a spin-up time. These oscillations are slightly correlated with the wind bursts lagging behind for about one day. Potentially, they are triggered by the remote forcing of CMEMS-HBM via the lateral boundaries which explains the time lag. Another cause could be vacillations of *KE* which is a well-known peculiarity of nonlinear rotating fluids (Hide, 1958; Früh,

10    2015). In order to filter out the oscillations, the cumulative average $\overline{KE_{R500\_NF}}$ was computed. This quantity is frequently used in time series analysis in order to determine the time scale at which a stochastic time series reaches stationarity. In the actual case, $\overline{KE_{R500\_NF}}$ 
[revised manuscript text omitted]

30    around $\pm 250$ m day$^{-1}$ (equivalent to $\approx 3$ mm s$^{-1}$) are associated with the above mentioned Class III textures.

In Section 4.2.1, it was already mentioned that the horizontal current patterns are somehow correlated with the water depth. Such a relationship applies as well for the patterns of the other variables displayed in Fig. 7. , e.g. the spatial variability of relative vorticity and the frequency of narrow filaments of confluent flow are clearly enhanced in the shallow water regions. An explanation for this

35    different behaviour is given by McWilliams (2019) who identified two principal mechanisms for the generation of STPPs:

(i) extraction of available potential energy due to horizontal gradients in the weakly stratified surface layer (mixed-layer instability), and (ii) topographic-drag vorticity generation in flows along a sloping bottom, followed by boundary current separation and wake instability. As strong bottom slopes are found only along Rönnebank and in the south of the R100 domain (see Fig. 1), the mechanism (ii) is obviously significant in those areas. However, for water depths > 50 m, the bottom is mostly flat and topographic-drag vorticity generation can be excluded.

[revised manuscript text omitted]
 < 0$ in those parts is primarily supported both by $Q_{geo} < 0$ and $Q_{dv} < 0$. A bimodal structure is also visible for F4. Here, frontogenesis is supported both by $Q_{geo}$ and $Q_{ageo}$, and frontolysis only by $Q_w$. For F3, the situation is similar.

The above results conflict to those obtained by Capet et al. (2008b), referred to as CMMS in the following: according to their Fig. 7, the "residual" (equivalent to $F$ in this paper) at the sea surface is generally negative and the geostrophic contribution is always positive, while the corresponding quantities in the current article exhibit a clear bimodal structure. On the other hand, the contributions from ageostrophic straining and vertical diffusion are bimodal in CMMS but $Q_{ageo}$ and $Q_{dv}$ are predominantly positive. Similarly, the impact from vertical straining in CMMS is everywhere negative, in contrast to $Q_w$ that is bimodal as well and the sign is alternating along-front. The cause for these disagreements might be the different horizontal resolution of ROMS (750 m in CMMS and 100 m in R100). By contrast, the multimodal structures and the alternating along-front sign changes of $F$ and its components resemble more those obtained by Gula et al. (2014).

Prominent structures of the tendency terms are also visible in C3 and C4, but those will be investigated in detail in a follow-up paper investigating the dynamics of a submesoscale cyclone.

**4.4.2 Submesoscale upwelling in eddies**

As noticed above, the vertical motion pattern in almost the entire model domain is impacted by internal waves that frequently blur the corresponding signals of fronts. However, in some settings, the vertical motion related to submesoscale fronts and eddies may supersede the internal wave-driven vertical velocity which can be seen in Fig. 7d. Such a situation is given during the life cycle of the submesoscale eddy C3 depicted in Fig. 11. C3 originates from a dense (cold and salty) streamer that invaded the area from the south, starting on 24 June around noon. While the streamer stretched farther to the north, it rolled up into C3 on 25 an 26 June (Fig. 12a–c; the entire process is shown in an animation on the web site https://www.hzg.de/institutes_platforms/coastal_research/operational_systems/coast_ocean_measurement/topics/index.php.de#tab-100, last access 1 March 2019). In Fig. 12d–f, the vertical speed at 5-m depth is shown for the same period of time. On 24 June before the winding-up of the eddy started, the vertical motion pattern is controlled by a train of internal waves travelling from northeast to southwest. One day later, the impact of the eddy formation on the vertical motion field becomes visible: two extended patches with vertical speeds of either sign are located south and north of the eddy core. On 26 June, the pronounced downwelling area is smaller but the upwelling still prevails at the same location. Apparently, the eddy-driven vertical motion supersedes the signal of the internal waves and causes persistent upwelling for a day or even more as can be seen in Fig. 12f: at the position of the crosshair, upwelling is still visible, although the eddy centre has moved already to the north by about 2 nmi.

**4.4.3 Instability mechanisms**

The results above impressively show that submesoscale eddies grow rapidly in the R100 domain, preferably in the shallow areas with water depths less than 40 m (Fig. 7). However, it is not yet clear if inertial, symmetric or mixed-layer instability drive the eddy growth. In the following, the R100 fields are analysed for criteria necessary for the occurrence of any type of these instabilities.

For inertial instability, negative absolute vorticity, $\zeta^a$, is a necessary condition, which is equivalent to $\zeta^a = \zeta + f < 0$ or $\zeta/f < -1$. It can be seen in Fig. 7b (blue regions) that $\zeta/f < -1$ is satisfied along the coast of Bornholm and at a few isolated locations along the meridional open boundaries, i.e. close to the centre of the cyclonic eddy in the northwest corner and between

54°25′ and 54°30′ N at the eastern boundary. However, the latter appearences are ignored because they are potentially due to false advection effects of relative vorticity discussed above. A zoomed image of $\zeta^a/f$ in the surface layer of R100 (Fig. 13) shows that negative values are found at 7 locations around Bornholm, indicating the birthplaces of disturbances driven by inertial instability. All of them are directly attached to the coast, but favourable conditions for the growth of the disturbances

5    seem to prevail only at sites no. 2 and 5 (yellow circles). In detail, on 26 June negative absolute vorticity is visible at no. 2, 3, 5, and 7. But within the subsequent 3 days, anticyclonic eddies have only developed at sites no. 2 and 5. It is common to 
[revised manuscript text omitted]

In R500, the kinematic and dynamical structures are rather sensitive to the surface boundary conditions. While the response of mesoscale patterns to the turning-off of the atmospheric forcing is rather sluggish, it has an immediate impact on the

generation of smaller-scale features which represent already the low-wavenumber part of submesoscale turbulence in the spectral range around 5 km.

In R100, the atmospheric forcing is turned-off from the outset because the air-sea fluxes inhibit the growth of STTPs. Thus, the R100 results represent situations that occur only under quasi-adiabatic conditions. In Nature, such situations are approximated in no-wind or light-wind conditions which offer the best chance to observe STPPs. On the other hand, the R100 findings must not be compared to observations which are taken during stronger wind. Moreover, as the R100 findings reflect summer conditions in the Baltic Sea, the must not be applied to another season or any other region of the World Ocean.

The horizontal density gradients in R100 grow for about 10 days, and afterwards the frequency of occurrence of strong gradients begins to decline, indicating frontal arrest as soon as the absolute horizontal density gradient reaches a critical value. STPPs develop rapidly within about a day; they are characterised by relative vorticities and divergences reaching multiple of the Coriolis parameter, where the frequency distribution of relative vorticity is clearly biased towards negative (anticyclonic) values. Vertical velocities of $\mathcal{O}(100)$ m day$^{-1}$ are diagnosed in R100. However, as the vertical motion is predominantly controlled by 
[revised manuscript text omitted]

Gurova, E., and Chubarenko, B. : Remote-sensing observations of coastal sub-mesoscale eddies in the south-eastern Baltic. Oceanologia, 54(4), 631–654, doi: 10.5697/oc.54-4.631, 2012.

Haine, T. W. N., and Marshall, J.: Gravitational, symmetric, and baroclinic instability of the ocean mixed layer. Journal of Physical Oceanography, 28, 634–658, 1998.

Haney, R. L.: On the pressure gradient force over steep topography in sigma coordinate models. Journal of Physical Oceanography, 21, 610–619, 1991.

Haidvogel, D. B., Arango, H. G., Hedstrøm, K., Beckmann, A., Malanotte-Rizzoli, P., and Shchepetkin, A. F.: Model evaluation experiments in the North Atlantic Basin: simulations in nonlinear terrain-following coordinates. Dynamics of Atmospheres and Oceans, 32, 239–281, 2000.

Hide, R.: An experimental study of thermal convection in a rotating liquid. Philosophical Transactions of the Royal Society of London (A), 250, 441–478, 1958.

Holton, J.: The role of gravity induced drag and diffusion in the momentum budget of the mesosphere. Journal of the Atmospheric Sciences, 39, 791–79, 1982.

Holton, J. R.: An introduction to dynamic meteorology. Fourth edition, Elsevier Academic Press, ISBN: 0-12-354016-X, 2004.

Hoskins, B. J.: The mathematical theory of frontogenesis. Annual Reviews of Fluid Mechanics, 14, 131–151, 1982.

Kantha, L. H., and Clayson, C. A.: An improved mixed-layer model for geophysical applications. Journal of Geophysical Research, 99, 25235–25266, 1994. Karimova2016

Karimova, S., and Gade, M.: Improved statistics of submesoscale eddies in the Baltic Sea retrieved from SAR imagery. International Journal of Remote Sensing, 37(19), 2394–2414, doi: 10.1080/01431161.2016.1145367, 2016.

[revised manuscript text omitted]

* as decribed by Onken (2017)

| Parameter | Units | Value | Meaning |
|---|---|---|---|
| $K$ | 1 | 10 | number of vertical layers |
| $V_{tr}$ | N/A | 2 | transformation equation |
| $V_{str}$ | N/A | 1 | stretching function |
| $\theta_s$ | 1 | 5 | surface control parameter |
| $\theta_b$ | 1 | 0.4 | bottom control parameter |
| $h_c$ | m | 10 | critical depth |
| $A_V^T$ | $m^2s^{-1}$ | $4 \times 10^{-5}$ | vertical mixing coefficient for tracers* |
| $A_V^M$ | $m^2s^{-1}$ | $1 \times 10^{-5}$ | vertical mixing coefficient for momentum* |
| $\Delta t_{fast}$ | 1 | 20 | Number of barotropic time-steps between each baroclinic time step |

**Table 2.** Parameter settings and properties of the R500 and R100 setups.

| Parameter/Property | Units | R500 | R100 | Meaning |
|---|---|---|---|---|
| $\Delta x$ | m | 500 | 100 | nominal zonal grid size |
| $\Delta y$ | m | 500 | 100 | nominal meridional grid size |
| $lon_{west}$ | | 13°30′ W | 14°12′ W | western boundary of model domain |
| $lon_{east}$ | | 16°30′ W | 15°48′ W | eastern boundary of model domain |
| $lat_{south}$ | | 53°54′ N | 54°18′ N | southern boundary of model domain |
| $lat_{north}$ | | 55°30′ N | 55°12′ N | northern boundary of model domain |
| $N_x$ | 1 | 386 | 1033 | number of tracer grid points (zonal) |
| $N_y$ | 1 | 356 | 1004 | number of tracer grid points (meridional) |
| domain size | $km^2$ | $193 \times 178$ | $103 \times 100$ | |
| $\Delta t$ | s | 150 | 30 | baroclinic time step |
| $A_H^T$ | $m^4s^{-1}$ | $5 \times 10^5$ | $10^3$ | bi-harmonic eddy diffusivity coefficient |
| $A_H^M$ | $m^4s^{-1}$ | $10^3$ | N/A | bi-harmonic eddy viscosity coefficient |
| $A_H^M$ | $m^2s^{-1}$ | N/A | $10^{-2}$ | mono-harmonic eddy viscosity coefficient |
| vertical mixing scheme | N/A | GLS | KPP | |
| $rx_0$ | 1 | 0.16 | 0.07 | stability condition after Haidvogel et al. (2000) |
| $rx_1$ | 1 | 1.75 | 0.84 | stability condition after Haney (1991) |

[Figure]

**Figure 1.** The R500 and R100 model domains in the western Baltic Sea. The approximate experimental area of the "Expedition Clockwork Ocean" is indicated by the dashed polygon.

[Figure]

**Figure 2.** Top-layer salinity in CMEMS-HBM (left column), R500 (centre), and R500_NF (right) for 1, 10, 20, and 30 June. The CMEMS-HBM fields are interpolated onto the R500 horizontal grid. Due to the coarser horizontal grid of CMEMS-HBM, inland lakes are not resolved and salinity values were assigned to places that are dry in R500. **31**

[Figure]

**Figure 3.** Instantaneous ($KE_{R500}$, $KE_{R500\_NF}$) and cumulatively averaged ($\overline{KE_{R500\_NF}}$) kinetic energy per unit mass of the model runs R500 and R500_NF. The wind speed is shown as black curve. All quantities are averaged over the model domain.

[Figure]

**Figure 4.** (a) Top-layer salinity of the R500 model on 23 June 09:00. (b) RGB composite from the Ocean and Land Colour Instrument of ESA satellite Sentinel-3 for the R500 domain on 23 June 09:32. The original image was manually adjusted in order to fit to the Mercator projection used in (a). C1, C2, and AC refer to the signatures of cyclonic and anticyclonis eddies, respectively. For the Oderbank, cf. Fig. 1.

[Figure]

**Figure 5.** R100: top-layer salinity (left column), potential temperature (centre), and the absolute horizontal gradient of potential density (right) on 20, 23, 26 and 29 June. The upper end of the colour axis of $|\nabla\rho|$ is limited to $4 \times 10^{-4}$ kg m$^{-4}$ to distinguish higher gradients from the dark background.

[Figure]

**Figure 6.** R100: frequency distribution of $|\nabla \rho|$ in the top layer. The initial and the open boundary conditions for R100 were provided by (a) R500 and (b) R500_NF, respectively. The values were binned in 10 intervals of $0.5 \times 10^{-3}$ kg m$^{-4}$ width between 0 and $5 \times 10^{-3}$ kg m$^{-4}$. Counts < 100 (gray shaded) representing less than 0.1% of the total number of horizontal grid points were not considered as significant.

[Figure]

**Figure 7.** R100, 26 June: (a) Magnitude and direction of top-layer horizontal velocity $\boldsymbol{V}$ (vectors are plotted at 3-km resolution), (b) relative vorticity $\zeta$ and (c) horizontal divergence $\delta$, each scaled by $f$, (d) vertical velocity $w$ at 5-m depth. The white line in (a) is the 40-m depth contour. The blue box in (a) refers to the zoomed area shown in Figs. 11 and 12, the box in (c) is the zoomed area of Fig. 8, and the box in (d) is the zoom shown in Fig. 15.

[Figure]

**Figure 8.** R100, 26 June: zoom of $f$-scaled horizontal divergence $\delta$ (grayscale image, for the position of the zoomed area see Fig. 7c). Bright areas indicate divergent flow, convergences appear dark. The magenta lines show the $|\nabla\rho| = 10^{-4}$ kg m$^{-4}$ contours. The yellow arrows likely mark internal wave packages as discussed in the text. The horizontal yellow line is the position of the section shown in Fig. 9. The dashed ruler represents a distance of 5 km

[Figure]

**Figure 9.** R100, 26 June: zonal section of vertical velocity $w$ at 54°49.9' N. Magenta contours indicate potential density anomaly $\sigma_\theta$, bold solid and dotted contours are isotachs of the meridonal velocity component $v$ [cm s$^{-1}$]. The position of the section is marked in Fig. 8.

[Figure]

**Figure 10.** R100 with full atmospheric forcing on 26 June: (a) Magnitude and direction of top-layer horizontal velocity $V$ (vectors are plotted at 3-km resolution), (b) relative vorticity $\zeta$ and (c) horizontal divergence $\delta$, each scaled by $f$, (d) vertical velocity $w$ at 5-m depth. For all subplots, the same color scaling was used as in Fig. 7.

[Figure]

**Figure 11.** R100: near-surface properties on 26 June within the blue box depicted in Fig. 7a. (a) Absolute horizontal density gradient $|\nabla\rho|$ $[10^{-4}$ kg m$^{-4}]$, and (b) scaled relative vorticity $\zeta/f$ in the top layer, (c) vertical velocity $w$ [m day$^{-1}$] at the base of the top layer. (d) – (i) The frontal tendency $F$ and the components of the tendency equation $Q_{geo}, Q_{ageo}, Q_w, Q_{dv}, Q_{dh}$ $[10^{-13}$ kg$^{-2}$ m$^{-8}$ s$^{-1}]$ at 2-m depth. Red marks frontogenesis, blue frontolysis. For the labels F1–F4, C3, C4 see text. The dashed rulers represent a distance of 2 km, vectors in (e) and (f) are drawn at 600-m intervals.

[Figure]

**Figure 12.** R100: (a)–(c) temporal evolution of potential density anomaly $\sigma_\theta$ in the top layer with vectors of total velocity $V$ drawn at 400-m resolution, (d)–(f) corresponding vertical velocity $w$ at 5-m depth; upwelling is positive. The area shown is indicated by the blue box in Fig. 7a. The crosshair tags the same position in all subplots. The dashed rulers represent a distance of 2 km.

[Figure]

**Figure 13.** R100: normalized absolute vorticity, $\zeta^a/f$, in the surface layer of the waters around Bornholm on 26 and 29 June. Encircled numbers indicate near-coastal locations of $\zeta^a/f < 0$. Yellow circles mark locations where inertial instabilities tend to grow offshore.

[Figure]

**Figure 14.** R100: (a, b) $\phi_{Ri}$, (c, d) $\phi_c$, and (e, f) $C_{SI}$ at 2-m depth on 15 (left column) and 26 June (right). (e) and (f) are logical maps indicating where the condition for symmetric instability, $C_{SI}$, is satisfied in regions of cyclonic (red) and anticyclonic (blue) absolute vorticity, according to eqs. (17) and (18), respectively. For explanation of the other symbols see text.

[Figure]

**Figure 15.** R100, 26 June: Potential density anomaly $\sigma_\theta$ in the top layer with total velocity **V** drawn at 600-m resolution. For the position of the area, see the red box in Fig. 7d. The black box indicates the position of the maps shown in Fig. 16a–e. The dashed ruler represents a distance of 2 km.

[Figure]

**Figure 16.** R100, 26 June: maps of dynamical quantities within the black box in Fig. 15. (a) $\sigma_\theta$ and $\mathbf{V}$, (b) $|\nabla\rho|$ and $\mathbf{V}$, (c) $\zeta/f$ and $\mathbf{V}_{geo}$, (d) $\delta/f$ and $\mathbf{V}_{ageo}$ in the top layer, and (e) $w$ and $\mathbf{V}_{ageo}$ at the bottom of the top layer. Velocity vectors are drawn at a resolution of 200 m. Vertical sections of (f) $v$, (g) $u_{ageo}$, (h) $\zeta/f$, (i) $\delta/f$, and (j) $w$. Contour lines of potential density anomaly $\sigma_\theta$ are magenta and spaced at the same intervals as in (a). The horizontal dashed lines in the centres of (a)–(e) indicate the position of the zonal sections displayed in (f)–(j). In either subplot, the dashed rulers represent a horizontal distance of 500 m.

[Figure]

**Figure 17.** R100, 26 June: Richardson number $Ri$ at 2-m depth and total top-layer velocity $\mathbf{V}$ drawn at 600-m resolution. For the position of the area, see the red box in Fig. 7d. The magenta box indicates the position of the maps shown in Fig. 16a–e. It is identical with the black box in Fig. 15. The dashed ruler represents a distance of 2 km.

[Figure]

**Figure 18.** Tracer patterns of submesoscale eddies. (a) Observed sea surface temperature of a cyclonic eddy in the Southern California Counter Current. The image was taken on 1 February 2013 at 20:34 in the framework of the SubEx experiment (Marmorino et al., 2018). (b) Modelled salinity and (c) potential temperature of the cyclone C3 on 26 June at 15-m depth (cf. Figs. 11 and 12). Cold water spots in (a) are denoted by "S". The dashed rulers in (b) and (c) represent a distance of 1 km.